# Carbon Dioxide Sensing—Biomedical Applications to Human Subjects

**DOI:** 10.3390/s22010188

**Published:** 2021-12-28

**Authors:** Emmanuel Dervieux, Michaël Théron, Wilfried Uhring

**Affiliations:** 1BiOSENCY, 1137a Avenue des Champs Blancs, 35510 Cesson-Sévigné, France; 2ORPHY, Université de Bretagne Occidentale, 6 Avenue Victor le Gorgeu, 29238 Brest, France; michael.theron@univ-brest.fr; 3ICube, University of Strasbourg and CNRS, 23 rue du Loess, CEDEX, 67037 Strasbourg, France; wilfried.uhring@unistra.fr

**Keywords:** carbon dioxide, CO_2_, transcutaneous monitoring, ptCO_2_, tcpCO_2_, paCO_2_

## Abstract

Carbon dioxide (CO${}_{2}$) monitoring in human subjects is of crucial importance in medical practice. Transcutaneous monitors based on the Stow-Severinghaus electrode make a good alternative to the painful and risky arterial “blood gases” sampling. Yet, such monitors are not only expensive, but also bulky and continuously drifting, requiring frequent recalibrations by trained medical staff. Aiming at finding alternatives, the full panel of CO${}_{2}$ measurement techniques is thoroughly reviewed. The physicochemical working principle of each sensing technique is given, as well as some typical merit criteria, advantages, and drawbacks. An overview of the main CO${}_{2}$ monitoring methods and sites routinely used in clinical practice is also provided, revealing their constraints and specificities. The reviewed CO${}_{2}$ sensing techniques are then evaluated in view of the latter clinical constraints and transcutaneous sensing coupled to a dye-based fluorescence CO${}_{2}$ sensing seems to offer the best potential for the development of a future non-invasive clinical CO${}_{2}$ monitor.

## 1. Introduction

In medical practice, the accurate monitoring of vital signs is of crucial importance to provide appropriate and effective care to the patients. In particular, the measurement of blood gases—namely di-oxygen (O${}_{2}$) and CO${}_{2}$—gives respiratory as well as circulatory clues on the state of a patient [1]. Yet, the continuous monitoring of the arterial partial pressure in O${}_{2}$–paO${}_{2}$—and CO${}_{2}$–paCO${}_{2}$—implies frequent arterial blood sampling, a process which is both painful and risky [2], requiring trained staff and expensive blood gas analyzers. Furthermore, the blood samples must be promptly analyzed upon collection, adding logistic constraints on the whole healthcare system [3]. Thus, the development of non-invasive paO${}_{2}$ and paCO${}_{2}$ monitoring techniques has been an active research field for decades [4,5,6], but while pulse oximetry proved to be a reliable proxy for paO${}_{2}$ [7,8], no satisfactory equivalent exists for paCO${}_{2}$.

On patients intubated or wearing a face mask, CO${}_{2}$ may be monitored in the exhaled breath, a technique called airway capnometry, leading to the measurement of the end tidal CO${}_{2}$ partial pressure, petCO${}_{2}$. Despite a poor correlation between petCO${}_{2}$ and paCO${}_{2}$ in case of ventilation-perfusion mismatch or elevated physiologic dead space, airway capnometry makes a good surrogate for blood gases measurement in case of stable haemodynamic conditions [9,10].

The other state-of-the-art paCO${}_{2}$ proxy is the transcutaneous partial CO${}_{2}$ pressure—tcpCO${}_{2}$: if the skin temperature is elevated enough to trigger reactive hyperaemia—and thus subcutaneous blood arterialization —tcpCO${}_{2}$ correlates well with paCO${}_{2}$ [11]. However, tcpCO${}_{2}$ monitors are bulky, expensive (∼15 k€), and—above all—need to be recalibrated at least every eight hours due to their important drift [12]. Consequently, there is a strong need for an alternative to the existing tcpCO${}_{2}$ monitors, with the aim of developing a cheap, non-invasive, stable, and accurate technique for long-term paCO${}_{2}$ monitoring.

Fortunately, since CO${}_{2}$ is ubiquitous in our world, its measurement has been the goal of many researchers from very different fields, each with its constraints and objectives. In particular, CO${}_{2}$ sensing is used: in the food storage and agri-food industry [13], in medical science [4], in sea and environmental research [14], and of course in the laboratories themselves (e.g., analytical chemistry). Such a diversity of applications leads to different constraints in terms of operating conditions (temperature and pressure), cost, durability, maintenance, etc.

In the first part, this article reviews existing CO${}_{2}$ sensing techniques available both on the market and in recent research. We aim at being exhaustive, not on the number of works gathered—we rely on external reviews on this point—but rather on the variety of techniques available. Our aim is to collect and present research works spanning the full range of CO${}_{2}$ sensing.

In a second part, the main goals and constraints intrinsic to biomedical monitoring are exposed. For each sensing medium—in blood or tissues, in exhaled breath, or at the skin surface—the clinical interest of its pCO${}_{2}$ is discussed. Then, the principal probing modalities routinely used in medical practice are detailed with their specificities, advantages, and drawbacks.

In a third part, a focus is made on transcutaneous sensing: past attempts to replace the ubiquitous Stow-Severinghaus electrode, specific constraints imposed by the transcutaneous sensing modality, and future direction for the development of a new kind of sensor. A brief conclusion summarising the key outcomes of this review is then drawn.

## 2. Review of **CO${}_{2}$** Sensing Techniques

### 2.1. Scope of the Review

As stated earlier, our aim is not to produce a complete bibliographic review of each mentioned technique. We do so for the sake of conciseness, and because comprehensive reviews focusing on one or another of the techniques presented here are available in the literature [15,16,17,18,19,20,21,22,23]. While the latter performs an in-depth coverage of their respective topic (e.g., nanomaterials-based sensors [18,22]), they never cover the full range of CO${}_{2}$ sensing possibilities. This is partly done, however, in the book *Carbon Dioxide Sensing* [24] which covers certain aspects of the present work in much more details, although letting others aside.

The present review spans the full range of CO${}_{2}$ measuring techniques with the exception of the analytical ones. That is for instance (mass) spectroscopy, magnetic resonance, chromatography, or titration. We did so because such techniques are not likely to be used for biomedical monitoring in the near future, due to the bulky and expensive apparatus they require. These methods are covered in-depth by dedicated analytical chemistry works, though [25,26].

For each technique, the underlying physical principle enabling CO${}_{2}$ measurement is exposed briefly. Such explanations are often accompanied by a clear figure or schematic to allow for rapid comprehension of the presented technique. The main features, advantages, and drawbacks of the technique are also disclosed, with typical characteristics of commercial or research applications, when available or relevant. The aim of such summaries is to provide the reader with a basic understanding of each exploited phenomenon, while giving them references to research articles or reviews to dig further. At the end of this section, a table is also given to summarise all the presented techniques and to compare them on several merit criteria—e.g., lifetime, accuracy, drift, cross-sensitivities, response time, form factor, etc.—see Section 2.6.

The structure of the review is organized around the physico-chemical properties of the CO${}_{2}$ molecule, namely:Infrared absorption of CO${}_{2}$—Section 2.2.Hydration of dissolved CO${}_{2}$ into carbonic acid—Section 2.3.Reduction of CO${}_{2}$ into CO${}_{2}^{-}$ and CO${}_{3}^{2-}$—Section 2.4.Acoustic properties of gaseous CO${}_{2}$—Section 2.5.

### 2.2. Infrared Absorption of CO${}_{2}$

#### 2.2.1. Non Dispersive Infra-Red (NDIR) Sensors

The operating principle of NDIR CO${}_{2}$ sensors is that of the Beer-Lambert law of absorption, for gaseous CO${}_{2}$ exhibits an absorbance peak at 4.26 µm, as can be seen in Figure 1. Due to the absence of other commonly encountered gases absorbing at this wavelength, NDIR sensors are very specific and can reach extremely low levels of detection if the sensing cavity is long enough. They operate as follows: an infrared source is placed on one end of a cavity containing the gaseous analyte, while an infrared receptor is placed at the other end. At a given wavelength λ, the measured light flux $\Phi_{mes}$ (W) is then function of: the emitted light flux $\Phi_{0}$ (W), the geometry of the sensor *k* (unit-less), the light path length *l* (m), the CO${}_{2}$ volume fraction $\chi_{{\mathrm{CO}}_{2}}$ (unit-less), and its absorbance $A_{{\mathrm{CO}}_{2}}$ (m${}^{-1}$) following [27]:(1)$$\Phi_{mes}\left(\lambda \right)=k\cdot \Phi_{0}\left(\lambda \right)\cdot e^{-l\cdot \chi_{{\mathrm{CO}}_{2}}\cdot A_{{\mathrm{CO}}_{2}}\left(\lambda \right)}$$

While this equation indicates a linear relationship between $log(\Phi_{mes}/\Phi_{0})$, *l*, and $\chi_{{\mathrm{CO}}_{2}}$, the actual linkage between the latter quantities is generally much more difficult to model accurately, due to a variety of light paths that contribute differently to the sensor response. Although this plurality of light paths can be studied beforehand by simulation—as did Hodgkinson, Liu et al. [29,30,31]—most sensors are calibrated empirically once manufactured.

Theoretically, only one emitter/receiver duo is needed, given one of them is narrow-band—or bandpass filtered—around 4.26 µm, so as to avoid interferences from other gases. In practice however, a *reference* channel is often placed beside the *measurement* one [32,33,34]. The role of this additional reference channel is to compensate for variations in the light source intensity due to temperature variations, for instance. In this case, the sensor is composed of two sensing elements, one is covered by a 4.3 µm bandpass filter, while the other one is covered by a bandpass filter in the 3.8–4.1 µm region, where no other gas absorbs. Other designs have also been proposed using a time-interleaved reference/measurement channel alternation, instead of two physically distinct light paths. In this case, a single sensing element is covered by a rotating filter wheel equipped with multiple bandpass filters at 4.2 µm and 3.8 µm [35]. The typical design of such referenced sensors is presented in Figure 2, Left.

NDIR sensors were often criticized because of the infrared source they used—i.e., a bulb with a heating filament—which both consumed much power and generated heat [36]. However, with recent advances in the domain of infrared emission and sensing, this is no longer a source of concern. In particular, recent InAsSb semi-conductors allows for reasonably cheap, energy-efficient emitters and receiver, e.g., AK9700AE (Light Emitting Diode (LED)) and AK9710 (sensor), AKM, Japan, or Lms43 LED and photodiode, LMSNT, Russia. Alternatively, thermopiles—possibly miniaturized as Micro Electro-Mechanical Systems (MEMS)—also offer good alternatives to the aforementioned photodiodes [37]. Similarly, MEMS emitters have also been developed [31,38].

The response time of NDIR sensors is limited by the gas flow rate in the sensing chamber, while their sensing range is a function of the light path used. Thus, it is possible to create fast sensors with operating range from a few hundred ppm up to 100% CO${}_{2}$ [36,39]. It should be noted, however, that since the detection of small concentrations requires a long light path—up to 80 mm for 100 ppm for instance [39]—it can lead to bulky sensors. In order to solve the latter issue, more complex sensing geometries can be envisioned to lengthen the light path while keeping a compact sensor, as can be seen in Figure 2, Right. Additionally, while NDIR CO${}_{2}$ sensors are not sensitive to relative humidity levels below 100% [35,36], condensation can lead to the formation of water droplets, either in the light path as a fog, or on the surface of light emitters, receivers, or reflectors, thus disturbing the measurements [39,40]. Fortunately, this latter issue may be solved by either detecting potential dew-point situations and removing potentially polluted measurements [41], or heating the sensor itself, though at the cost of a higher power consumption [42,43].

Reviews of mid-infrared sources [44], NDIR applications for gas sensing in general [32,45], and concrete applications to CO${}_{2}$ measurement [30,33,39,46] can be found in the literature. Interestingly, even if predominantly reported in the gas phase, NDIR CO${}_{2}$ measurements may also be performed onto an aqueous solution containing dissolved CO${}_{2}$ [47].

#### 2.2.2. Photoacoustic Sensors

Photoacoustic sensors also use the afore-mentioned absorbance of CO${}_{2}$ in the mid-infrared (4.26 µm) and periodically illuminate the CO${}_{2}$ present in the sensing volume. Either a mechanical chopper with a continuous source, a low-inertia source, or even a pulsed laser is used to produce a periodic infrared illumination on the gas sample to analyze. If CO${}_{2}$ is present in the gas mixture, it absorbs the infrared radiation and thus heat, dilating slightly. When the illumination is stopped, the mixture cools down and thus compresses. The alternating illumination causes a repetition of these dilatations and compressions, which is nothing more than an acoustic wave. The latter can in turn be measured with a microphone [48].

This technique requires a light source that can be modulated at high frequencies—depending on the geometry of the cell and the sensor used—which is achievable using Light Emitting Diodes (LEDs) or lasers, but precludes the use of a light bulb as an infrared light source—unless a mechanical chopper is used, of course. Depending on the targeted accuracy or measurement range, several designs may be employed. The most simple one consists of a non-resonant acoustic cell with a simple microphone and a modulated light working in the 20 Hz–20 kHz range. However, using a resonant cell design and placing the microphone at an antinode of the acoustic wave can lead to much higher output levels. Such a design is illustrated in Figure 3 with an organ-pipe-like resonant cell. With cells in the centimeter range, this translates into frequencies below 40 kHz. Further still, the microphone can consist of a quartz tuning fork—a piezoelectric transducer with a quality factor above 10,000.

With the use of resonant photoacoustic cells or microphones—e.g., quartz tuning forks—very high sensitivities of a few ppm or even ppb can be achieved [49]. Yet, quartz tuning forks are influenced by temperature and humidity—even if this cross-sensitivity is reproducible and can thus be compensated for—and may need to be frequently recalibrated [50]. Contrariwise, MEMS microphones appear to be independent of humidity—even if being temperature-sensitive—and no frequent calibration need was reported [51]. Abundant examples of photoacoustic sensors can be found in the literature [51,52,53] and recent research is ongoing, targeting their miniaturization into MEMS sensors [54]. Besides, photoacoustic CO${}_{2}$ sensors can be designed for the full range of CO${}_{2}$ sensing—from a few ppb up to 100% CO${}_{2}$—and their response time is mainly limited by the gas flow inside the sensor. It should be noted that this flow is necessarily limited since high flow value generates turbulences, which are essentially acoustic noise [48].

### 2.3. Hydration of CO${}_{2}$ into Carbonic Acid

When presented to an aqueous medium, gaseous CO${}_{2}$ dissolves into carbonic acid (H${}_{2}$CO${}_{3(aq)}$), which further dissociates into bicarbonate (HCO${}_{3(aq)}^{-}$) and carbonate (CO${}_{3(aq)}^{2-}$) ions, following [55,56]:(2)$${\mathrm{CO}}_{2(aq)}+H_{2}O\overset{K_{1}}{⇌}H_{2}{\mathrm{CO}}_{3(aq)}\;\left\{\begin{array}{c} K_{1}=1.5\times 10^{-3}\\ pK_{1}=2.8 \end{array}\right.$$
(3)$$H_{2}O+H_{2}{\mathrm{CO}}_{3(aq)}\overset{K_{2}}{⇌}H_{3}O_{\left(aq\right)}^{+}+{\mathrm{HCO}}_{3(aq)}^{-}\;\left\{\begin{array}{c} K_{2}=4.44\times 10^{-7}\\ pK_{2}=6.35 \end{array}\right.$$
(4)$$H_{2}O+{\mathrm{HCO}}_{3(aq)}^{-}\overset{K_{3}}{⇌}H_{3}O_{\left(aq\right)}^{+}+{\mathrm{CO}}_{3(aq)}^{2-}\;\left\{\begin{array}{c} K_{3}=4.67\times 10^{-11}\\ pK_{3}=10.33 \end{array}\right.$$
wherein typical values of $K_{i}$ can be found in the literature [57,58] and are given here in pure water at 25 °C. The consequences of such dissociations are twofold: first, CO${}_{2}$ dissolution tends to lower the pH of the aqueous medium, and second, the presence of dissolved ions induces changes in the conductance of the solution. While the former phenomenon is the basis the Stow-Severinghaus electrode, Ion-Selective Field-Effect Transistor (ISFET) sensors and dye-based sensors, the former one drives the operation of conductometric CO${}_{2}$ sensors.

#### 2.3.1. Wet Conductometric Sensors

Wet conductometric sensors take advantage of the fact that when CO${}_{2}$ dissolves in an aqueous solution, it generates both HCO${}_{3}^{-}$ and H${}_{3}$O${}^{+}$ ions, which in turn modifies the conductivity of the solution [59]. A typical static conductometric sensor is composed of a chamber filled with distilled water and containing two electrodes. The chamber is then covered with a CO${}_{2}$ permeable membrane to allow outer CO${}_{2}$ to diffuse inside the sensor [60,61,62]. Alternatively, a dynamic conductometric sensor may be built, into which distilled water is circulated between a gas diffusion area and an impedance measurement area [13,59]. A static measurement cell is depicted in Figure 4. Using such an apparatus, the sensing medium may be either liquid or gaseous, while dynamic flow-through geometries have only been reported using gaseous analytes.

Such sensors are in theory inexpensive, easy to build, and stable. In practice however, a drift is often observed whose roots are not accurately known, even if some authors suggest contamination of the distilled water with external ions. In particular, if glues or resins are used to attach the membrane to the sensor, or to assemble the different parts of the embodiment together, they may further suffer from outgassing and release ions in the distilled water [61]. If the membrane is not tightly sealed, foreign ions may also enter the solution and thus change its conductivity. Drift correction [63] or automatic recalibration [64] techniques have thus been developed to compensate for this phenomenon. Alternatively, other authors immersed the body of their electrode for at least two weeks in bi-distilled water to make sure that foreign ions possibly bound to the glass surface re-dissolve in the rinsing water and are thus eliminated [65]. The dynamic flow-through geometry also eludes this issue by forcing the water into a de-mineralizing resin while being recycled.

As is the case for other membrane-covered sensors—see Section 2.3.2 below, for instance—the water can still evaporate through the covering membrane depending on the ambient relative humidity, especially in the case of static sensors used on a gaseous analyte. This issue was partially addressed by Neethirajan et al. [66], who presented a polytetrafluoroethylene (PTFE) membrane-covered sensor using Nafion as a proton conductor over a layer of polyaniline boronic acid, which exhibits a conductivity change upon a change in its pH. Overall, wet conductometric CO${}_{2}$ sensor can be used on the full 0–100% CO${}_{2}$ range, with response times below 1 min. Still, their life has not been reported in the literature and may be seriously impacted by the afore-mentioned drift.

#### 2.3.2. The Stow-Severinghaus Electrode

The Stow-Severinghaus electrode—named after its inventors Richard Stow [4] and John Severinghaus [67]—is the current state-of-the-art measurement method for transcutaneous pCO${}_{2}$ [11]. Even if it has been miniaturized since its first introduction, its basic working principle remains unchanged. As one can see in Figure 5 it is nothing more than a pH-meter plunging in an electrolyte and covered with a thin membrane. Originally designed for ex vivo blood pCO${}_{2}$ sensing, it was later modified to be used transcutaneously as well [68].

Dissolved CO${}_{2}$ diffuses from the outer medium into the sensor’s embodiment through a membrane made of an ion-impermeable, CO${}_{2}$-permeable material. A rubber membrane was originally proposed but was quickly replaced by a PTFE one [4,67], although silicone rubber is still occasionally used for its ease of application and better adhesion [69]. The electrolyte was also switched from distilled water to a solution of sodium carbonate (NaHCO${}_{3}$) and sodium chloride (NaCl) in various concentrations, which influences both the sensitivity and response time of the electrode [55]. CO${}_{2}$ dissociation lowers the pH of the electrolyte—as described by Equations (Equation 2)–(Equation 4)—which is then measured using a pair of electrodes: the glass electrode, and the reference electrode. The glass electrode consists in an Ag/AgCl electrode more often than not, although a Sb/SbO${}_{x}$ electrode has also been occasionally reported [68], while the reference electrode can be made of Pt [67,70] or AgCl [68]. Alternatives to the glass electrode have been explored with liquid membranes instead of the traditional glass bulb [71], or a replacement of the whole electrode by an iridium oxide film grown electrochemically on an iridium wire, yielding a higher sensitivity and being easier to manufacture than the glass electrode [72,73].

Apart from miniaturization and the exploration of other membrane materials, redox electrode pairs, or pH membrane material, the electrode underwent little changes compared to its original design. The response time of the Stow-Severinghaus electrode is generally reported to lie in the 1–5 min range, although the addition of carbonic anhydrase—a catalyst to the reaction (Equation 2)—led to response times below 1 min [69,74]. Still, the long-term stability of the electrode rarely exceeds a few months and some amount of drift is always present, making it necessary to re-calibrate the electrode every few hours or days, depending on the target accuracy [71,75]. The full range of the Stow-Severinghaus electrode is usually that of physiological pCO${}_{2}$ range—that is ∼2.6–13.2 kPa (or ∼20–100 mmHg) [11]—although applications for concentrations as low as ∼0.05–0.1% have also been reported [71,76]. The temperature dependency of the electrode may be compensated for, and a mixture of water and ethylene glycol can be used as a base for the electrolyte, preventing the electrode from drying out from a loss of water vapor through the gas-permeable membrane [77].

#### 2.3.3. ISFET Sensors

Due to the limited miniaturization potential of the Stow-Severinghaus electrode, Ion-Selective Field-Effect Transistors (ISFETs) were explored as CO${}_{2}$ sensing devices. ISFETs are Metal Oxide Semiconductor Field-Effect Transistor (MOSFET) whose gate is $H_{3}O^{+}$ sensitive. The addition of a reference electrode can turn them into fully functional pH meters. The use of ISFET for pH sensing has been reported as early as 1970 [78] and was later extended to CO${}_{2}$ sensing with the addition of a layer of bicarbonate solution covered with a gas-permeable, ion-impermeable, silicone rubber membrane [79].

As can be seen in Figure 6, the ISFET gate insulator is not metalized but rather covered with a layer of silicon oxide (SiO${}_{2}$), itself optionally covered with another layer of silicon nitride (Si${}_{3}$N${}_{4}$), hafnium oxide (HfO${}_{2}$), aluminum oxide (Al${}_{2}$O${}_{3}$) or tantalum pentoxide (Ta${}_{2}$O${}_{5}$). Once hydrated, the upper layer of the gate insulator becomes able to exchange protons with the surrounding electrolyte, thus acting as a pH electrode. Still, another electrode—the reference electrode—is needed to apply or measure a grid-source potential on the transistor. This latter potential—$V_{GS}$—itself function of the pH of the electrolyte, modulates the conductance of the transistor, hence turning it into a pH sensor [80]. CO${}_{2}$ detection can then be performed by covering the electrolyte with a CO${}_{2}$-permeable membrane, impermeable to other ions. This basic approach has been successfully applied by a number of authors in the literature, either for pH measurements [79,81], or direct CO${}_{2}$ sensing [14,82,83,84].

Among the above-mentioned gate insulators, hafnium oxide offers better performances in terms of drain current, sensitivity, and body effect than silicon nitride. Hafnium oxide may even be used alone, in place of the silicon oxide layer [85]. The inner electrolyte usually consists of a bicarbonate solution, while the reference electrode is often an AgCl/Ag electrode. ISFET CO${}_{2}$ sensors are subject to the same drawbacks as the Stow-Severinghaus electrode, namely the drift of the reference electrode and temperature dependency, although the latter may be compensated for. An approach for drift correction has also been proposed with good results either on restrained [81] or wide pH ranges [86]. The dry-out and humidity sensibility problems have been addressed with a dry sensor [87], even if the drift of the latter has yet to be reported. Alternatively, a low-drift design with a deported reference electrode has also been proposed with promising results [88].

#### 2.3.4. Dye-Based Sensors

Dye-based CO${}_{2}$ sensors also rely on the influence of CO${}_{2}$ hydration and dissociation on the pH of an aqueous solution. The pH change caused by the dissociation of CO${}_{2}$ translates into a change in the optical properties of a pH-sensitive dye, which can exist in either a protonated—RH—or an anionic—$R^{-}$—form. The dye is chosen so that its protonated and anionic forms differ either in their absorbance or luminescence spectra, and also for having a $K_{A}$ acidity constant near that of reaction (Equation 3) ($pK_{A}=-log\left(K_{A}\right)\approx 6.3$ in water at 25 °C [89]). It can be either dissolved in an aqueous solution—which is in turn entrapped between a substrate layer and a gas-permeable, ion-impermeable membrane—or dissolved in a thin polymer film with the aid of a phase transfer agent. These two kinds of sensors are often referred to as “wet” and “dry” sensors, respectively. A typical fluorescent sensor is depicted in Figure 7, along with the fluorescence spectrum of 1-hydroxy-pyrene-3,6,8-trisulfonate (HPTS), a commonly encountered pH-sensitive fluorophore.

The most basic sensors use a single wavelength for illumination—in case of absorbance measurements—or a pair of excitation-emission wavelengths—in case of luminescence measurement. Photobleaching may be compensated for, using more wavelengths and ratiometric methods similar to that driving pulse oximetry [90,91,92,93,94,95,96,97], while variations in the emitted light intensities or sensitivities toward received light can be addressed by using referencing—i.e., taking a measurement in pure di-nitrogen (N${}_{2}$) and/or pure CO${}_{2}$ prior to CO${}_{2}$ measurement for sensor calibration purposes [98,99,100,101,102,103,104,105,106,107,108,109,110,111,112,113,114].

Three other more sophisticated sensing schemes have been developed through the years, in addition to the aforementioned techniques, using an additional pH-independent, reference fluorescent dye: namely the Inner Filter Effect (IFE), Dual Lifetime Referencing (DLR), and Fluorescent Resonance Energy Transfer (FRET). Those techniques are presented in more detail in literature reviews or research articles focusing either on dye-based sensors in general [56,115] or on specific applications such as the IFE [116,117,118,119,120,121,122,123], DLR measurements [124,125,126,127,128,129,130] or FRET [131,132,133,134]. Surprisingly, none of these methods appears to be notably better than another, with some state-of-the-art works using either DLR [130], IFE [122] or simple referencing [113].

In contrast, a clear distinction exists between dry and wet sensors, with dry sensors largely outperforming wet ones, would it be in terms of response time, long-term stability, or ease of manufacturing [56,135]. Dry sensors are basically made out of a substrate layer, onto which a polymer is cast. The dye(s) of interest is prior dissolved into the polymer through the mean of a phase transfer agent before casting. A plasticiser is also often added to enhance the CO${}_{2}$ diffusion inside the polymer layer. Optionally, the latter may be covered by an additional protective layer, so as to prevent ionic contamination, or destruction of the sensor by acidic vapors. Apart from the optical sensing scheme, the main differences between the many dry sensors mentioned in the literature lie in the choice of the substances used for the aforementioned layers. Although no exhaustive comparison can be found, several authors compared different combinations of chemicals:**Substrate:** the substrate material does not seem to have any major influence on the sensing performance and is indifferently made of polyethylene terephthalate (PET) films, glass slides, or optical fibers. Nonetheless, polyethylene naphthalate (PEN) appears to be superior to PET, because its lower diffusivity towards CO${}_{2}$ yields shorter response times [128]. In addition, the adhesion of the polymer film on the glass substrate can be difficult without a cumbersome surface preparation [112,127] while PEN foils may just be roughened to enhance the sensitive layer adhesion [128].**Polymer:** hydroxy propyl methylcellulose (HPMC) may be a better choice than ethyl cellulose for being more hydrophilic than the latter, allowing water molecules to be entrapped with the dye and favoring the hydration of CO${}_{2}$ into bicarbonate ions [121].**Plasticiser:** several plasticizers were compared in the literature, tributyl phosphate (TPB) and Tween 20 appear to be the most efficient ones in terms of sensor response time [105,121].**Phase transfer agent:** cetyltrimethyl ammonium hydroxide (CTAH) tends to lead to shorter response times [125] although tetramethyl ammonium hydroxide (TMAH) may be more stable since it is less susceptible to Hofmann elimination [121].**Covering membrane:** Hyflon or even Cytop if a minimal response time is not mandatory may be used to limit the sensor poisoning and humidity loss [128].

Overall, dye-based sensors cover the full 0–100% CO${}_{2}$ range with response times below 1 min—a few seconds have even been seldom reported [110,113,119]—accuracies below 1%, lifetimes approaching one year [122,123] and thicknesses as low as 1 µm [112]. Still, they may be destroyed quickly by acidic vapors, and often suffer from a high cross-sensitivity towards humidity, O${}_{2}$—for it is a fluorescence quencher [136,137]—and temperature. While a protective membrane may mitigate the first two drawbacks, temperature influence may be compensated for, and O${}_{2}$ cross-sensitivity can be reduced, either by using polymers with a low oxygen permeability [120,122] or by fixing the O${}_{2}$-sensitive luminophore on polystyrene micro- or nano-beads [125,127,138]. Using a dual-frequency DLR scheme may also be used to compensate for O${}_{2}$ luminescence quenching [124].

#### 2.3.5. Optical Fiber Sensors

Although optical fibers have been used in some of the afore-mentioned dye-based sensors to convey light back and forth between bulky bench-top equipment and compact sensing membranes or solutions [92,94,101], they did not consist in the CO${}_{2}$-sensitive part itself. On the contrary, optical fibers-based CO${}_{2}$ sensors put optical fibers at the very heart of their sensing principle, using one of the four different techniques presented in Figure 8. These four sensing schemes can be grouped in two categories: either (i) the CO${}_{2}$ sensitive material changes the total light path length by shrinking or dilating in response to a change in CO${}_{2}$ concentration and thus turns the optical fiber into a miniaturized Fabry-Perrot interferometer (case (b)), or (ii) the CO${}_{2}$ sensitive material exhibits a change in its refractive index or exert a mechanical constraint on the fiber core. In these last two cases, changes are made to the evanescent modes of the fiber, introducing a shift in the transmitted or reflected wavelengths (cases (a), (c), and (d)).

The principle behind end-of-fiber Fabry-Perrot interferometer-based sensors is to measure the interferences between the light wave reflected by the fiber-film interface and that reflected by the film-analyte interface. Depending on the wavelength of the probing light and the thickness of the film, the interferences can be either constructive or destructive, causing notches and peaks in the observed reflected spectrum. The exact positions of those spectral features are directly linked to the thickness of the film, and the latter fluctuates with the surrounding pCO${}_{2}$. Thus, accurate spectral measurements of the shift in wavelength of a given notch can yield reliable pCO${}_{2}$ measurements [139,140].

As for evanescent modes-based fibers, several sensing modalities exist. Fiber Bragg gratings may be used as a strain gauge on a block of pH-sensitive hydrogel that swells or contract with pH changes, thus bending the fiber it is attached to [141]. Applying a mechanical strain on the optical fiber modifies its effective refractive index [142], inducing a wavelength shift in the evanescent modes of the grating. This technique does not require an optical fiber in itself, however, since a regular pressure sensor may be used instead, possibly much cheaper and more convenient to implement [143]. The pH-sensitive hydrogel can also directly coat the optical fiber optic, with similar consequences, as exerting strain on it [144]. Other materials, which exhibit a direct change in their refractive index upon CO${}_{2}$ adsorption, have also been explored for optical fiber coatings —such as metal-organic frameworks or ionic liquids—with some success [20,145].

Alternatively, other optical fiber-based sensors have been developed relying on principles closer to NDIR absorption than CO${}_{2}$ hydration. Those utilize the absorption of evanescent waves by CO${}_{2}$ present in the air surrounding the fiber at ∼1.5 µm, where CO${}_{2}$ absorbs. An additional coating of the fiber was shown to greatly improve the sensitivity of such sensors towards CO${}_{2}$ [146,147].

The main drawback of optical fiber-based sensors, apart from the price of the associated measurement apparatus, is their cross-sensitivity towards external strain, temperature, relative humidity, and possibly other chemicals which can be absorbed by the optical fiber coating. Those sensors exhibit response time in the order of 1 min on the full 0–100% range. To our knowledge, the accuracy of such CO${}_{2}$ sensors was never characterized extensively, although Melo et al. [146] measured a 4.1% resolution on the full scale. Of note, if multiple sensors are used in the same facilities or geographical area, multiplexing can be used to share the light source and measurement apparatus between them, thus greatly reducing the cost per sensor [148]. Such multiplexing was demonstrated for probing up to 256 sensors on the same fiber, and distances as high as 150 km were reported [149].

### 2.4. Reduction of CO${}_{2}$

Sensors based on the reduction of CO${}_{2}$ into CO${}_{2}^{-}$ and CO${}_{3}^{2-}$ radicals operate following several schemes. CO${}_{2}$ may be reduced following its adsorption on a thin film of metal oxide, thus modifying the conductivity of the oxide layer—Section 2.4.1. A similar adsorption mechanism has also been reported between CO${}_{2}$ molecules and graphene, leading to graphene-based sensors—Section 2.4.2. Alternatively, the reduction of CO${}_{2}$ may be used in an electrochemical cell, whose electromotive force yields the surrounding CO${}_{2}$ activity—Section 2.4.3. Lastly, ionic liquids have been used in CO${}_{2}$ sensing, either binding to CO${}_{2}$ to form carbamate species in conductometric sensor or acting as a substrate for CO${}_{2}$ reduction into peroxydicarbonate ions in amperometric sensors—Section 2.4.4.

#### 2.4.1. Adsorption by Metal Oxide Thin Film

Adsorption-based sensors use the change in conductivity of a thin layer of metal oxide caused by the adsorption of CO${}_{2}$. Although the latter adsorption has been studied for a number of years [150,151,152,153,154], its mechanism is still not entirely understood and remains a discussed topic in the scientific community [155,156,157,158]. Indeed, both physical adsorption and chemisorption appear to take place at the metal-oxide interface, and the reduction of CO${}_{2}$ into CO${}_{2}^{-}$ may not be the only mechanism involved [157]. That being said, the following reaction is often considered in the absence of a more satisfactory alternative [159,160,161]:(5)$$\begin{array}{cc} {\mathrm{CO}}_{2(gas)}+e^{-} & ⇋{\mathrm{CO}}_{2(ads)}^{-}\\ {\mathrm{CO}}_{2(ads)}^{-}+O_{\left(lat\right)}^{-} & ⇋{\mathrm{CO}}_{3(ads)}^{2-} \end{array}$$
wherein *ads* stands for *adsorbed*, *lat* for *lattice*, and the O${}^{-}$ radicals and $e^{-}$ are supposed to come from the MO lattice—where M may be Zn or Cu, for instance. Overall, despite the use of metal oxide CO${}_{2}$ sensors by an important number of authors, few of them propose an explicit sensing scheme, and even fewer mind using equilibrated or justified schemes, which may lead to incoherences. For instance, it has been proposed that since the above-mentioned scheme consumes free electrons from the MO lattice, fewer of them remain available for conductivity [162]. The resulting expected behavior would be an increase in the metal oxide resistance with an increase in pCO${}_{2}$. While this assertion seems to hold in the simple ZnO case [160,162,163], opposite behaviors were observed when doping ZnO with lanthanum [164] or calcium/aluminium [165,166]. Consequently, we believe that further research is needed to gain a deeper insight into the working principle of thin-film metal oxide sensors. Concerning the present review, we are nevertheless able to mention some interesting works and put them into perspective.

The main drawbacks of most metal oxide sensors are their elevated operating tempera ture—in the 200–600 °C range—and cross-sensitivity towards carbon monoxide (CO), O${}_{2}$, H${}_{2}$, and humidity. Along the years, several metals were explored, the most ubiquitous being copper, zinc and—although to a lesser extent—cadmium and tin, but chromium (Cr${}_{2}$O${}_{3}$) [153], barium/tungsten (Ba${}_{x}$WO${}_{y}$) [167], barium titanate (BaTiO${}_{3}$) [163] and lanthanum (La${}_{2}$O${}_{2}$CO${}_{3}$) [168] compounds were also studied. While the performance of chromium-based sensors was not measured, La${}_{2}$O${}_{2}$CO${}_{3}$-based sensors exhibit a fairly good sensitivity but a long response time—over 1 h—and Ba${}_{x}$WO${}_{y}$-based sensors showed an important drift. BaTiO${}_{3}$-based sensors were loaded with different metal oxides, and promising results were obtained with zirconium and lanthanum oxides, with response times below 10 min, and almost no humidity influence near 600 °C.

Several elements were used as doping agents for CuO or ZnO thin film such as the above mentioned lanthanum and calcium/aluminium, but sodium [169], germanium/neodymium/tungsten [160], manganese [170], gold [161], silver [155,171], and indium [172] were also explored. While the manganese-, gold-, and indium-doped ZnO-based sensors reported in the literature performed worse than their pure ZnO counterpart, all the other above-mentioned doping elements greatly enhanced the sensitivity of the sensors—more than doubling it in certain cases [166].

Overall, metal oxide sensors can only work at an elevated temperature—above 100 °C at least, even if 200–600 °C are more usual. Rare attempts to operate at room temperature resulted in sensors that appeared to be rather unstable and highly sensitive to humidity [173,174], or were not thoroughly characterised [161,169]. Sensitivity to humidity may be mitigated by the addition of silver in the metal oxide [163,175], while some degree of CO and H${}_{2}$ cross-sensitivity seems to be inevitable at first glance.

Yet, a temperature modulation approach—a.k.a. temperature cycling approach—may be used to isolate the sensor response due to CO${}_{2}$ from that of other interfering gases. This approach has been first suggested in the early 1980s [176], and was quickly demonstrated in the following years as a proof-of-concept [177,178,179,180,181]. In brief, temperature modulation is performed by cycling the operating temperature of a metal oxide sensor, and recording its dynamic response. Since both (i) the peak sensitivity as a function of temperature and (ii) the kinetics of adsorption/desorption onto the metal oxide layer vary from one gas to another, it is possible to discriminate between different gases using the same sensor operated at different cycling temperatures [182]. This approach has been successfully used to discriminate between organic—e.g., benzene, methane, ethanol [180,181]—and inorganic—e.g., CO, H${}_{2}$S [178]—gases, or mixture thereof. Both qualitative—i.e., classification of an unknown gas mixture in pre-determined categories [183,184]—and quantitative—i.e., accurately measuring the concentration of each gas in a given mixture [185,186]—have been performed successfully. Applications to CO${}_{2}$ sensing can also be envisioned, in spite of more scarce literature on the topic [187,188].

Measurements in the physiological range have been reported [163] and response times below 30 s are reachable [160,171,189]. The influence of humidity has also been discussed in detail by Gankanda et al. [159] and the above-mentioned hydration of CO${}_{2}$ is suspected to alter the sensing properties of metal oxide sensors. Interestingly, while most works focused on the change in conductivity of metal oxide upon CO${}_{2}$ adsorption, other works were based on capacitive [171,175,190], work function [173,174] or optic properties [161] measurements. Thorough reviews on metal oxide-based sensors for gas sensing—although not focused on CO${}_{2}$ detection alone—are also available [23,191].

#### 2.4.2. Adsorption by Graphene

More recently, a similar approach has been used with graphene as the sensing material instead of a metal oxide film [192,193] and—as is the case for metal oxide-based sensor—the working principle of such sensors is also a debated topic. Overall, graphene acts as a room-temperature p-type semiconductor and can be used either in a purely conductometric setup—measuring the bulk resistance or impedance of the graphene layer [194,195,196]—or in a Field-Effect Transistor (FET) setup—of which graphene is the canal [197]. However, the interaction of the CO${}_{2}$ molecule with the graphene layer is not exactly known, and different competing theories have been proposed. On the one hand, CO${}_{2}$ has been presented as an oxidizing or reducing agent, which exchanges electrons with the graphene lattice, the direction of the charge transfer depending on the applied electric field [198]. On the other hand, it has also been proposed that the change of conductivity of the graphene layer following CO${}_{2}$ adsorption could be caused by adsorption-induced impurity scattering without formal charge exchange [199].

Graphene has been used both as a pristine material [194,200,201,202] and in its oxidised form [196,203] for CO${}_{2}$ sensing. Additionally, other authors even considered the fabrication of composite sensors, combining graphene with polyethyleneimine [195] or metal oxides—such as Al${}_{2}$O${}_{3}$ [204], Sb${}_{2}$O${}_{3}$ [205], In${}_{2}$O${}_{3}$ and NiO [206] with mitigated results. Indeed, the performance of graphene-based devices for CO${}_{2}$ sensing suffers from a high cross-sensitivity to a variety of other gases—e.g., N${}_{2}$, O${}_{2}$, NH${}_{3}$, CO or H${}_{2}$ [196,200,201]—and humidity [201,202,207]. Encouragingly, however, Akther et al. [196] recently reported temperature and humidity corrections for graphene oxide-based CO${}_{2}$ sensor, with a mean error below 3% on the full 400–4000 ppm scale, and a response time of 3–5 s, despite showing some degree of cross-sensitivity towards other gases. So as to overcome the latter cross-sensitivity issue in graphene-based sensors, Rumyantsev et al. [208] developed a discrimination method based on the analysis of the low-frequency noise spectrum of a pristine graphene transistor. However—at least to the best of our knowledge—their method is yet to be applied to CO${}_{2}$ sensing.

#### 2.4.3. Electrochemical Cells

Potentiometric sensors based on electrochemical cells have been proposed for CO${}_{2}$ sensing since the late seventies [209], and although several ameliorations have been developed in half a century of intense research, their basic working principle remains largely untouched. An extensive description of the sensing principle of such sensors for CO${}_{2}$ can be found in the work of Maruyama et al. [210] and will be briefly summarised here. The sensor itself consists of the following electrochemical cell:(6)$$\mathrm{Au},{\mathrm{CO}}_{2},O_{2}|{\mathrm{Na}}_{2}{\mathrm{CO}}_{3}\left|\right|{\mathrm{Na}}_{2}O|O_{2},\mathrm{Au}$$
wherein the ion bridge is made of NASICON (Na${}_{3}$Zr${}_{2}$Si${}_{2}$PO${}_{1}2$), which acts as a sodium-ion conductor. The anode, cathode, and global cell reactions are given by:(7)$$\begin{array}{ccc} 2\cdot {\mathrm{Na}}^{+}+{\mathrm{CO}}_{2}+\frac{1}{2}\cdot O_{2}+2\cdot e^{-} & ={\mathrm{Na}}_{2}{\mathrm{CO}}_{3}\; & \left(\mathrm{anode}\right)\\ 2\cdot {\mathrm{Na}}^{+}+\frac{1}{2}\cdot O_{2}+2\cdot e^{-} & ={\mathrm{Na}}_{2}O\; & \left(\mathrm{cathode}\right)\\ {\mathrm{CO}}_{2}+{\mathrm{Na}}_{2}O & ⇋{\mathrm{Na}}_{2}{\mathrm{CO}}_{3}\; & \left(\mathrm{global}\right) \end{array}$$
and the electromotive force of the cell appears to be proportional to log(pCO${}_{2}$). The anode, covered in sodium carbonate (Na${}_{2}$CO${}_{3}$) is referred to as the *sensing electrode*, while the cathode is the *reference electrode*. The outline schematic of such a potentiometric sensor is presented in Figure 9.

A major improvement to the stability of potentiometric sensors has been the addition of a third reference electrode [210,211,212], or the use of a solid reference electrode instead of the above-mentioned Na${}_{2}$O/O${}_{2}$ interface [213,214,215]. Still, even with this referencing, such sensors usually suffer from some degree of cross-sensitivity towards humidity. In addition, one of their major drawback for biomedical applications is their high operating temperature—in the 300–700 °C range.

The performance of a potentiometric sensor strongly depends on the thickness of its different constituents, but also on the choice of the materials used for the electrolyte and the sensing electrode coating. Concerning the electrodes themselves, their composition does not seem to have any significant influence on the sensing performance, since they are often made out of noble metals—such as gold [210,211,216] or platinum [209,212,217]. The electrode wires are bound to the solid electrolyte using pastes of the same metal—possibly deposited with a stencil—which are further sintered to make an electrical bond between the solid electrolyte and the wires. For a wider contact area, the use of a gold mesh has also been proposed [216]. The solid electrolyte consists more often than not in NASICON, but potassium carbonate (K${}_{2}$CO${}_{3}$) [209], Na-beta-alumina [218], lithium phosphate (Li${}_{3}$PO${}_{4}$) [213,214,215], and more recently Yttrium-doped LSBO (La${}_{9.66}$Si${}_{5.3}$B${}_{0.7}$O${}_{26.14}$) [219] and Li${}_{7}$La${}_{3}$Zr${}_{2}$O${}_{1}2$ [220] have also been used with some success. As for the coating of the sensing electrode, various carbonates were explored such as Na${}_{2}$CO${}_{3}$, Li${}_{2}$CO${}_{3}$ or CaCO${}_{3}$, with similar performance. Reference materials such as yttrium stabilised zirconia [210], BiCuVO${}_{x}$-perovskite-oxide [211], Li${}_{2}$TiO${}_{3}$/TiO${}_{2}$ [213,214,215] and Bi${}_{8}$Nb${}_{2}$O${}_{17}$ [212] were used exhibiting good stability.

If the objective comparison of one of the combinations of the above-mentioned chemicals with another is made difficult by the change in form-factor and film thicknesses of the reported works, one can still notice some remarkable facts. At first, in an attempt to mitigate the humidity cross-sensitivity of potentiometric sensors, the addition of barium carbonate to a lithium carbonate sensing material (BaCO${}_{3}$/Li${}_{2}$CO${}_{3}$) [211,213,215], or the addition of lithium carbonate to an indium tin oxide sensing material (Li${}_{2}$CO${}_{3}$/ITO) [216] have been successfully explored by several authors, yielding potentiometric sensors with no cross-sensitivity toward humidity. Then, room temperature operation of such sensors was also achieved with good sensitivity at 30 °C on the 300–3000 ppm CO${}_{2}$ range [216]. Finally, thin-film designs were investigated yielding particularly fast—below 1 min—responding sensors [214,215,221]. The full range of potentiometric CO${}_{2}$ sensors covers the 0–100% CO${}_{2}$ concentration.

#### 2.4.4. Ionic Liquids-Based Sensors

Room-temperature ionic liquids have been intensively studied since the early 2000s, with targeted applications for gas sensing and CO${}_{2}$ sequestration in particular [222,223,224,225]. Ionic liquids, unlike ordinary liquids, are not made out of neutral molecules but consist exclusively of ion pairs. They also exhibit a negligible vapor pressure and their electrical conductivity makes them appropriate candidates to act as liquid electrolytes. As for metal oxide sensors, the exact mechanisms underlying CO${}_{2}$ sensing with ionic liquids are still partially unknown, although three main pathways have been proposed. One is the reduction of CO${}_{2}$ at the anode and its further oxidation at the cathode, possibly following [226]:(8)$${\mathrm{CO}}_{2(ads)}+e^{-}⇋{\mathrm{CO}}_{2(ads)}^{-}$$
although—since there is a reasonable amount of doubt about the latter equation and the existence of CO${}_{2(ads)}^{-}$—most authors did not explicit it, or preferred the R-CO${}_{2}$ notation—standing for *Reduced*-CO${}_{2}$—instead of CO${}_{2}^{-}$ [227]. CO${}_{2}$ reduction can be measured either by a change in the impedance of the ionic liquid [228,229,230,231,232], or by amperometric or cyclic voltammetry techniques [226,227,233].

A second sensing scheme has been proposed, in which CO${}_{2}$ binds with an amine to form a Carbamate Ionic Liquid (CIL) following [234]:(9)$$\begin{array}{cc} \mathrm{RNH}+{\mathrm{CO}}_{2} & ⇋\mathrm{RNCOOH}\\ \mathrm{RNCOOH}+\mathrm{RNH} & ⇋\underset{\mathrm{CIL}}{\underset{︸}{{\mathrm{RNH}}_{2}^{+}+{\mathrm{RNCOO}}^{-}}} \end{array}$$

Since the CIL conducts electricity, it can serve as a CO${}_{2}$ probe using different sensing schemes. Although a simple impedance measurement may be performed, Chen et al. [234] exploited the chemoluminescence of luminol in the presence of O${}_{2}$ instead. In their sensor, the presence of CO${}_{2}$ in solution generated a conducting CIL as described by Equation (Equation 9). Electric current is then passed through the CIL, reducing dissolved O${}_{2}$ into O${}_{2}^{-}$ ions which oxidizes luminol. Oxidized luminol in turn emits light by chemiluminescence, which can be quantified by optical measurements.

A third sensing scheme takes advantage of the irreversible reduction of CO${}_{2}$ into peroxydicarbonate in the presence of oxygen following [235,236]:(10)$$\begin{array}{cc} O_{2}^{•-}+{\mathrm{CO}}_{2} & \to {\mathrm{CO}}_{4}^{•-}\\ {\mathrm{CO}}_{4}^{•-}+{\mathrm{CO}}_{2} & \to C_{2}O_{6}^{•-}\\ C_{2}O_{6}^{•-}+O_{2}^{•-} & \to C_{2}O_{6}^{2-}+O_{2} \end{array}$$

Since it is not reversible, the latter scheme may not be used for a traditional CO${}_{2}$ sensor, but rather for CO${}_{2}$ detection, dosimetry, or sequestration. Interestingly, this reduction of CO${}_{2}$ in the presence of oxygen has been treated as interference by several authors [226,233], who placed an oxygen trap before their CO${}_{2}$-sensitive ionic liquid layer to prevent it from happening. This reduction may also be the cause of the slight drift observed in some of the afore-mentioned, non-oxygen-protected sensors [228,231,232].

In most cases, ionic liquids sensors are probed by amperometric or conductimetric methods, such as cyclic voltammetry, galvanostatic, or impedance measurements. However, Fabry-Perot interferometry—see Section 2.3.5—has also been reported [237] as well as acoustic resonant sensors using quartz micro-balances, for instance [236,238]. The idea of such sensors is to assess the shift in frequency of a resonant element coated with an ionic liquid. When the ionic liquid adsorbs CO${}_{2}$, its mass increases, lowering the resonance frequency of the micro-balance. When CO${}_{2}$ desorbs, the inverse phenomenon occurs. Of note, such acoustic resonant sensors based on CO${}_{2}$ solubility were also explored with silicone or amine-based polymers instead of ionic liquids [239,240,241,242,243], although they often exhibit significant sensitivity to humidity.

Overall, ionic liquid sensors have been used on the physiological [226] or full 0–100% [227,238] CO${}_{2}$ range. Response time in the 40–200 s seems reachable [226,234,238] although much longer response times have also been reported, especially for CO${}_{2}$ desorption when returning to baseline [230,231,232]. The accuracy of such sensors was seldom characterized, with only Chen et al. [234] mentioning a 2.05% accuracy measured on 10 trials. The same remark holds for the sensor lifetime, reported in only one case to be over 4 months [226]. Ionic liquids sensors also exhibit a temperature-dependent response [226,233], and are often sensitive to CO, O${}_{2}$, H${}_{2}$, NO${}_{2}$, N${}_{2}$O, SO${}_{2}$ and humidity [226,228,234].

### 2.5. Acoustic Properties of CO${}_{2}$

#### 2.5.1. Time of Flight Acoustic Sensors

Time of flight sensors are based on the difference in sound velocity between CO${}_{2}$ and other gases—e.g., 349, 326 and 267 m·s${}^{-1}$ for N${}_{2}$, O${}_{2}$ and CO${}_{2}$ respectively [244]. Thus, a simple time of flight sensor can be constructed for CO${}_{2}$ sensing, consisting of an ultrasound emitter and receiver pair, facing each other in the analyte gas mixture. Joos et al. [245] performed acoustic measurements using a 20 cm tube at the end of which the ultrasound transducers were attached. The authors were able to measure the CO${}_{2}$ content of a CO${}_{2}$/N${}_{2}$ gas mixture on the full 0–100% CO${}_{2}$ range with an accuracy of 0.3%. An outline schematic representing the working principle of their experiment is depicted in Figure 10.

While this technique may seem interesting at first glance for its low cost and long-term stability, it is usually limited to a mixture of two gases [246]. Consequently, it might be unusable in the context of biomedical CO${}_{2}$ monitoring. Indeed, most human tissues not only produce CO${}_{2}$ and consume O${}_{2}$ but also evaporate water. As such, a gaseous volume equilibrated with human tissues may contain as many as four different gases—CO${}_{2}$, O${}_{2}$, N${}_{2}$ and H${}_{2}$O—whose amounts cannot be found with a single pulse velocity measurement. These sensors usually have long path lengths—20 cm in the above-mentioned work [245]—which may be difficult to embed in a portable apparatus. Still, they can provide low latency measurements, since their response time is only limited by the time taken by the analyte gas mixture to fill the acoustic chamber. Interestingly, recent research by Cicek et al. [247] demonstrates another exploitation of the difference in sound velocity between N${}_{2}$ and CO${}_{2}$, using an array of miniature Helmholtz resonators, whose resonance frequency shifts depending on the CO${}_{2}$ concentration to which they are exposed. They achieved CO${}_{2}$ sensing in the 0–1% range with a 70 mm radius sensing array.

#### 2.5.2. Acoustic Attenuation Sensors

The acoustic attenuation sensing scheme is based on the vibrational relaxation phenomenon. Vibrational relaxation consists of the attenuation in the intensity of a propagating acoustic wave in a gas. This attenuation depends on the sound frequency and also on the gas—or gas mixture—in which it takes place [248,249]. Acoustic attenuation measurements performed at different frequencies allow establishing the acoustical absorption spectrum—i.e., attenuation intensity as a function of the acoustic wave frequency—of a gas mixture. Several such spectra are presented in Figure 11 and can be used to identify the different gases that compose an analyte gas mixture as well as their proportion in the latter. To do so, an apparatus similar to that presented in Figure 10 may be used, but with multiple emitter/receiver pairs placed at different distances. Then, measurements of the sound attenuation and velocity can be performed, yielding an absorption value. The latter may in turn be compared with theoretical calculations—or experimental data—for a CO${}_{2}$/N${}_{2}$ gas mixture, leading to a pCO${}_{2}$ value.

Theoretical and practical applications of this phenomenon are available in the literature [250,251,252,253,254,255]. However, at least to the best of our knowledge, vibrational relaxation applied to CO${}_{2}$ sensing never went further than the proof of concept presented by Petculescu et al. [254]. Contrary to other sensors presented in this review, cross-sensitivity towards O${}_{2}$ should be negligible. Indeed, O${}_{2}$ does not exhibit any vibrational relaxation phenomenon in the 10${}^{3}$–10${}^{7}$ Hz·atm${}^{-1}$ [253]. The influence of humidity may be another source of concern. If the lower mode of water vibrational relaxation occurs at much lower frequencies than that of CO${}_{2}$ [256], higher frequencies components are also present at high humidity fraction which can interfere with CO${}_{2}$ measurements [253]. In addition, the temperature should be measured and compensated for, as it influences sound propagation, and thus the position and amplitude of the acoustical absorption peaks [251].

### 2.6. Comparison Table

Although an objective comparison of the above-mentioned CO${}_{2}$ measurement techniques is made exceedingly difficult due to the incompleteness of most of the referenced works, we tried to do our best to gather the state-of-the-art performance for the different sensing schemes presented. The pith and marrow of our literature review are thus summarised in Table 1.

We gently warn the reader that each column of the table contains the best performance reported in the literature for a given criterion and that no sensor may exist with all the characteristics presented in a given row. For instance, concerning ionic liquids-based sensors, Chen et al. [234] reported a 2.05% accuracy but with a 2 min response time, while Mineo et al. [238] reported a 40 s response time without reporting the accuracy of their sensor. Still the “ionic liquid” row of Table 1 contains both the 2.05% accuracy and 40 s response time, even if no sensor exists with such characteristics.

Finally, although most criteria are self-explanatory, a few additional comments may be necessary for a sound understanding and interpretation of this table. First, the lifetimes of several sensor families—and especially that of the (photo-)acoustical ones—are not explicitly reported in the literature. Yet, since NDIR sensors were reported to reach a lifetime of at least 15 years [36], and since acoustical building blocks—e.g., ultrasonic transducers—usually last for years, the infinity symbol (*∞*) was used, meaning that a lifetime of at least several years should be expected. Second, the calibration need refers to the need to re-calibrate a sensor frequently—i.e., every few hours or days—due to the presence of a significant drift in its response. This criterion does *not* refer to the initial sensor calibration that may be performed in the factory. Third, the response time of the evaluated sensors was often measured using test benches which flush the sensors with a continuous flow of a gas mixture containing a known CO${}_{2}$ concentration. In such a configuration, the four sensors noted with an (a) are supposed to respond as soon as the latter gas mixture flows inside their embodiment, hence the ≤1 s indication. In practice, however, the response time strongly depends on the sensor housing dimensions, since CO${}_{2}$ will have to diffuse from the analyte gas or medium inside the sensor embodiment.

## 3. Current Applications to Biomedical **CO${}_{2}$** Sensing

As a by-product of aerobic metabolism, CO${}_{2}$ is ubiquitous in the human body and its influence on pH—see Section 2.3—makes its regulation in the organism a key parameter to maintain homeostasis [263]. In particular, CO${}_{2}$ equilibrium is controlled by several mechanisms governing the efficiency of gas exchange [264,265] and by renal regulation of plasma bicarbonate ions [266]. Both the excess or deficiency of CO${}_{2}$ in the organism—termed hyper- and hypocapnia, respectively—can have either beneficial or deleterious effects on a patient’s outcome, depending on their condition [267]. As a consequence, CO${}_{2}$ monitoring is of major importance in clinical care. Its measurement can be performed in three locations, presented in order of decreasing invasiveness:inside the body with blood or tissues sampling—Section 3.1,in the exhaled air with airway capnometry—Section 3.2,on the skin with transcutaneous capnometry—Section 3.3.

An illustration of these locations is provided in Figure 12. For each one of them, the clinical significance of the corresponding pCO${}_{2}$ measurement is first presented, and its probing modalities in practice are then described. One should bear in mind through the remainder of this section, that CO${}_{2}$ transport in the human body takes three distinct forms—dissolved CO${}_{2}$, bicarbonate ions, and carbamate compounds—which are in constant equilibrium for a given blood pH, and could give rise to as many measurements—blood pCO${}_{2}$, [HCO${}_{3}^{-}$], Carbamino-Haemoglobin (CO2 Hb) concentration and pH [268,269]. Yet, for both practical and clinical reasons, pCO${}_{2}$ measurements have been predominantly adopted from a clinical perspective, as is disclosed below.

### 3.1. Blood Gas Analysis

#### 3.1.1. Clinical Significance

Arterial blood sampling is considered to be the gold standard for the assessment of whole-body CO${}_{2}$ content. Arterial blood is released from the lungs and is thus fully oxygenated, with a low CO${}_{2}$ content, which is not the case of capillary or venous blood. Arterial blood pCO${}_{2}$—paCO${}_{2}$—thus best represents the global haemodynamic status of patients and gives important clues on their metabolism and homeostasis [1,270]. There is an extremely large panel of clinical applications of paCO${}_{2}$ measurements including acute and chronic respiratory failures [271,272], mechanical ventilation assessment [273,274], general anaesthesia—but also sedation in patients at risk of hypoventilation—monitoring [275], or resuscitation procedures [276]. Due to the painful and potentially risky aspects of arterial blood puncture [2], other blood sampling techniques have been explored. In particular, arterialized capillary blood sampling was reported to be an acceptable surrogate for arterial blood sampling [277]. In contrast, peripheral venous blood can not be used to this end [278], nor can central venous blood [279,280,281], even if both may be used for trend analysis. Venous blood pCO${}_{2}$ measurement is nevertheless interesting in critically ill patients for whom measuring the venous-to-arterial carbon dioxide partial pressure difference (p_v-a_CO${}_{2}$)—or calculating the venous-to-arterial carbon dioxide content difference—makes it possible to detect organ hypo-perfusion [282].

More localized pCO${}_{2}$ measurements have also been performed with intra-tissue probing—in particular into the brain [283], liver [284] or skeletal muscles [285]—yielding crucial information about the perfusion of the organ under study. Gastric tonometry and sublingual capnometry have also been explored, but their clinical interest is yet to be fully demonstrated [286,287].

#### 3.1.2. Probing Modalities

In vivo CO${}_{2}$ probing can be achieved under two distinct modalities. Either pCO${}_{2}$ is measured in situ or a biological sample is collected to be further analyzed.

In the first case, the sensor is brought to the analyte. This is notably what happens for intra-tissue pCO${}_{2}$ monitoring. In such cases, the sensor is inserted directly into the organ or blood vessel that is to be probed by mean of a catheter, for instance. This setting allows for continuous monitoring with a low latency, which is only dictated by the response time of the chosen pCO${}_{2}$ sensor [288].

In the second case, often used for blood pCO${}_{2}$ measurements, the analyte is brought to an external analyzer and an arterial or venous line placement may be performed to facilitate repeated blood sampling. However, these settings allow only discrete monitoring since a blood sampling must be performed every single time a pCO${}_{2}$ is desired. In addition, the blood samples must be analyzed quickly upon collection, which may lead to additional logistic difficulties compared to in situ sensing [3], even if recent hand-held blood gas analyzers tend to mitigate this issue [289].

In practice, two main technologies have been used to perform blood gas analysis: optodes using dye-based CO${}_{2}$ sensing as described in Section 2.3.4 [288,290] and electrochemical sensors such as the Stow-Severinghaus electrode, described in Section 2.3.2 [291].

### 3.2. Airway Capnometry

#### 3.2.1. Clinical Significance

Airway capnometry—from Greek *capnos* (καπνoς), smoke, vapor —is the measurement of the amount of CO${}_{2}$ in exhaled air, with a distinction between capnometry—which consists in the measurement of petCO${}_{2}$, the end-tidal pCO${}_{2}$—and capnography—which usually refers to the plot of a capnogram: pCO${}_{2}$ in the exhaled air as a function of time, or volume. The distinction between capnometry and capnography is made clear in Figure 13.

Capnography is particularly useful in medical practice since it gives information about respiratory airflow, CO${}_{2}$ production, and elimination, respiratory quotient, or the quality of endotracheal intubation. It can also be used to detect ventilation/perfusion inadequacy, apnea, Chronic Obstructive Pulmonary Disease (COPD), or heart failures [294,295]. For its part, petCO${}_{2}$ has proven to be a reliable proxy for paCO${}_{2}$ under stable haemodynamic conditions, with petCO${}_{2}$ being usually 2–5 mmHg lower than paCO${}_{2}$ [294]. However, the correlation between petCO${}_{2}$ and paCO${}_{2}$ vanishes in case of elevated physiological dead space or ventilation-perfusion mismatch [1,296]. It can also be difficult to use capnometry on neonates due to the small volume of exhaled air that they produce [297].

Interestingly, Siobal et al. [294] pointed out that while airway capnometry is often criticized for its inaccuracy as a paCO${}_{2}$ proxy—e.g., in case of ventilation/perfusion mismatch, elevated dead space, or poor endotracheal placement—the same roots of this inaccuracy may be identified and quantified by a simultaneous paCO${}_{2}$ measurement. In other words, if simultaneous paCO${}_{2}$ and petCO${}_{2}$ measurements are performed in the same patient, discrepancies between the two values may reveal one of the aforementioned issues.

#### 3.2.2. Probing Modalities

Two distinct modalities exist for airway capnometry: mainstream or sidestream measurements. In mainstream capnometry, the CO${}_{2}$ sensor is positioned on the main airway of the intubated patient so that the whole breathed air flow is forced through it. This technique yields instantaneous petCO${}_{2}$ measurement and real-time capnogram plot. Historically, mainstream capnometry has long been criticized, for the additional sensor on the patient’s airway was bulky, fragile, and led to an additional respiratory dead space. In particular, the bulkiness of mainstream sensors may dislodge the endotracheal tube in young patients [294,297]. In response, recent designs have been improved to become more rugged, compact, and with minimal sampling volume, partially mitigating these drawbacks [298,299].

In contrast, sidestream capnometry consists of a small diameter sampling tube, which continuously aspirates a fraction of the patient’s breathed air. The air is then conveyed to an external monitor wherein both the analysis and display functions are performed. Contrary to the mainstream technique, a delay is present between air sampling and the monitoring of its CO${}_{2}$ content [292]. Additionally, distortions of the measured capnogram may occur, especially on its more rapidly changing portions, and sidestream-measured petCO${}_{2}$ may be slightly underestimated compared to the mainstream-measured one [300].

In practice, airway capnometry is more often than not measured using the infrared absorption of CO${}_{2}$, either using NDIR sensors—as presented in Section 2.2.1—or photoacoustic sensors—as presented in Section 2.2.2 [292,294]. Additionally, colorimetric dye-based sensors—as described in Section 2.3.4—are also routinely used to assess the correct positioning of endotracheal intubation, although they only provide qualitative information about the latter, and not a quantitative tcpCO${}_{2}$ reading [301].

### 3.3. Transcutaneous CO${}_{2}$ Sensing

#### 3.3.1. Clinical Significance

Transcutaneous pCO${}_{2}$–tcpCO${}_{2}$—measurements are clinically relevant in two different situations: either as a surrogate for paCO${}_{2}$ in patients with normal tissue perfusion or as an evaluation tool to measure the paCO${}_{2}$–tcpCO${}_{2}$ gap in patients with an abnormal perfusion [302].

In patients with normal circulation and peripheral perfusion, tcpCO${}_{2}$ correlates well with paCO${}_{2}$ and, as such, may be used in all situations where paCO${}_{2}$ is required [11]. This correlation is particularly influenced by the choice of the measuring site, and by the probe temperature, as will be disclosed in the next section.

Alternatively, measurements of the paCO${}_{2}$–tcpCO${}_{2}$ gap proved to be a reliable predictor of mortality in case of shock [303]. In such a view, however, a simultaneous measurement of paCO${}_{2}$ is required in addition to tcpCO${}_{2}$ monitoring. One may notice the similarity between this technique, and the proposal of Siobal et al. for petCO${}_{2}$ mentioned earlier—see Section 3.2.1.

#### 3.3.2. Probing Modalities

tcpCO${}_{2}$ measurements can be performed with the attachment of a transcutaneous probe on the skin by means of a disposable, adhesive mounting ring. The measuring site and sensor temperature depend on what is to be measured: for instance, when searching for a paCO${}_{2}$ surrogate, it is recommended to place the tcpCO${}_{2}$ probe at the earlobe with a setpoint temperature above 42 °C for the best results [11]. However, an elevated skin temperature can be a source of skin burn or thermal injury, especially in the neonates, requiring a frequent change of the probe site. A compromise may then be found between skin temperature—and thus site change frequency—and accuracy of the tcpCO${}_{2}$ measurement [304]. When assessing local perfusion or shock condition—on the contrary—a temperature as low as 37 °C may be used at the site of interest [305]. Different temperatures can also be used to assess changes in perfusion as a function of temperature [303].

Technically speaking, the tcpCO${}_{2}$ is measured by means of a miniaturized Stow-Severinghaus electrode similar to that described in Section 2.3.2 which can be heated anywhere in the 37–45 °C range [302].

## 4. Future Applications to Transcutaneous Monitoring

While the above-mentioned monitors are routinely used in clinical practice, they are not only expensive and invasive but also unpractical—for they require frequent external interventions for calibration or sampling purposes—and bulky, as mentioned in Section 1 and Section 3. These drawbacks, in addition to turning pCO${}_{2}$ monitors into a coveted resource used sparingly in the hospital, make them unusable outside the hospital. Yet, bringing pCO${}_{2}$ monitors in homes would allow for telemonitoring, which can reduce hospital admissions and the risk of contracting nosocomial infections. Although literature is inexistent regarding pCO${}_{2}$ telemonitoring for the obvious reason that an adapted monitor does not exist yet, several clinical trials are encouraging regarding the advantages of telemedicine—a.k.a. telehealth—on patient’s outcome and costs of admission in various pathologies [306,307,308]. Further still, both the emergence of new pandemics such as the COVID-19 outbreak [309] and the rapid growth of health wearables [310,311,312,313] may also push medical practice towards telemonitoring.

In this context, the development of a non-invasive, low-cost, and portable—if not wearable—biomedical pCO${}_{2}$ monitor appears highly desirable. Among the three sensing modalities presented in Section 3, only transcutaneous pCO${}_{2}$ monitoring meets the non-invasiveness target underlined above. In the following sections, the rare attempts found in the literature at finding an alternative to the drifting Stow-Severinghaus electrode conventionally used for tcpCO${}_{2}$ monitoring are first given. Then, our recommendations regarding the development of a future sensor meeting the above-mentioned criteria follows, taking into account the technological review presented in Section 2.

### 4.1. Past Attempts

The Stow-Severinghaus electrode appeared in the late 1950s, underwent only minor improvements since then, and reigns supreme over the tcpCO${}_{2}$ monitoring market [4,302]. Yet, several authors have been trying to push forward other means to measure tcpCO${}_{2}$ mainly using NDIR sensing and mass spectroscopy, as disclosed in the following sections. It is also worth mentioning that we measured the absorption spectrum of CO${}_{2}$Hb—i.e., haemoglobin bound to CO${}_{2}$—while searching for a non-invasive alternative to tcpCO${}_{2}$ monitoring. In a nutshell, we wondered whether the optical properties of CO${}_{2}$Hb could pave the way for an in vivo pCO${}_{2}$ measurement technique analogous to pulse oximetry but for CO${}_{2}$. Alas, this track proved to be but a dead-end, as CO${}_{2}$Hb, and Deoxy-Haemoglobin (HHb) have identical absorption spectra [314].

#### 4.1.1. Non-Dispersive Infrared and Transcutaneous Sensing

Regarding NDIR, a couple of works by Eletr, Greenspan et al. surfaced in the early 1980s [315,316], describing a Hewlett-Packard capnometer which collects CO${}_{2}$ exhaled by the skin in a small sampling chamber probed by infrared radiations. Despite the fair performances achieved—up to 48 h of continuous monitoring, a STD of 0.2 kPa (1.5 mmHg) between paCO${}_{2}$ and tcpCO${}_{2}$ on 3–6 h sessions and a response time about 90 s [315]—the main drawback of this sensor was the need to strip the stratum corneum with 20–50 successive applications of adhesive tape so as to reach a reasonable response time.

The NDIR track was then dropped until the mid-2010s and the advent of rate-based approaches by the team of Chatterjee, Ge, Iitani et al. [317,318,319,320] at first and by the team of Grangeat et al. [321,322] in a second move. If the latter work is insufficiently detailed—especially the explanations concerning the sensing part itself—the former allows for a sound understanding of the rate-based approach. This approach consists in placing a sampling cup against the skin to collect transcutaneously exhaled CO${}_{2}$, but—contrary to what was done in the earlier work of Eletr, Greenspan, et al.—no time is left for the gas in the cup to reach an equilibrium with the skin CO${}_{2}$ content. Rather, the gas in the cup are flushed with fresh air or pure N${}_{2}$ and its CO${}_{2}$ fraction measured by a conventional NDIR sensor. Theoretically, the so-obtained CO${}_{2}$ diffusion rate is only function of (i) the patient’s tcpCO${}_{2}$ and (ii) their skin diffusivity towards CO${}_{2}$ [318]. However, since the latter parameter is not known accurately and fluctuates a lot depending on the skin temperature [323,324], measurement site [325,326] and subject at hand [327], the rate-based approach is most likely limited to trend analysis at best.

In the afore-mentioned works, it should also be noted that contrary to the teams of Chatterjee, Ge, Iitani, et al., those of Grangeat, Eletr, Greenspan, et al. heated the skin in the 35–43 °C range.

#### 4.1.2. Mass Spectroscopy and Transcutaneous Sensing

Overcoming the issue of the latter rate-based approach, a more sophisticated technique has been explored in parallel by Mc Ilroy et al. in the early 1980s [328,329,330] using mass spectrometry for CO${}_{2}$ detection. The main idea behind their work is to perform a per-subject and per-site calibration to first determine the skin permeability towards CO${}_{2}$ and only then access to the patient’s tcpCO${}_{2}$. While in its first version, their sensor required a change of membrane to perform this calibration [329], a second version with two sampling chambers and an additional valve solved this need for external intervention [330]. In their setup, the skin was also stripped with 5–10 applications of adhesive tape beforehand, and heated to 43–45 °C. They achieved a correlation factor of 0.81 between paCO${}_{2}$ and tcpCO${}_{2}$, and a response time below 5 min. The main drawback of their technique is of course the need for a mass spectrometer, which is bulky, requires maintenance, and costs tens if not hundreds of thousands of euros. Their dual-rate technique, however, might presumably be adapted to an NDIR sensing scheme, thus compensating for the fluctuations of skin diffusivity mentioned in the previous section.

### 4.2. Transcutaneous Sensing Constraints

An effective biomedical tcpCO${}_{2}$ sensor should fulfill two main objectives: it must respond swiftly to a patient’s haemodynamic changes, and it must measure a tcpCO${}_{2}$ that reflects accurately their state. To do so, the sensor design must take into account the two main characteristics of CO${}_{2}$ diffusion through the skin, which both are temperature-dependent:The correlation between the pCO${}_{2}$ measured at the skin surface—that is tcpCO${}_{2}$—and that of arterial blood—that is paCO${}_{2}$, which will determine the ability of the sensor to give clues about the patient’s status through a tcpCO${}_{2}$ reading.The exhalation rate of CO${}_{2}$ through the skin into the sensor, which will determine the response time of the sensor.

#### 4.2.1. Correlation between paCO${}_{2}$ and tcpCO${}_{2}$

The latter correlation has been demonstrated at 42 °C and above [11], but heating the skin is not only uncomfortable for the patient but also energy-consuming—which limits the ability of the sensor to be integrated into a wearable, battery-powered device—and potentially harmful due to risk of thermal injury.

Indeed, literature on the topic suggests a maximum long-term skin temperature of 42 °C, even if higher temperatures can be used for a limited amount of time—up to 45 °C for a few minutes for instance [331], or up to several hours depending on the study [332]. In the medical practice of tcpCO${}_{2}$ monitoring, temperatures up to 45 °C were used for up to one hour [333] or 42 °C for eight hours [12]. Legally speaking, the Food and Drug Administration (FDA) guidance for tcpCO${}_{2}$ monitors recommends not to exceed 44 °C for more than four hours [334].

For these reasons, working at low skin temperature seems highly desirable. To the best of our knowledge, only the work of Wimberley et al. [333] studied the influence of skin temperature on the correlation between paCO${}_{2}$—-through arterialized capillary blood, to be accurate—and tcpCO${}_{2}$. Their study covers the full 37–45 °C range and they show a fair—r=0.93—correlation between capillary and transcutaneous pCO${}_{2}$, even at a temperature as low as 37 °C. Yet, further research on the topic would be valuable since other sources argue that a higher probe temperature still yields narrower limits of agreement for tcpCO${}_{2}$ [11].

#### 4.2.2. Exhalation Rate

The *exhalation rate* of CO${}_{2}$ through the skin refers here to the macroscopic observation of cutaneous respiration. It corresponds to a volumetric flow rate of gaseous CO${}_{2}$ per unit area—dimension L${}^{1}$·T${}^{-1}$, unit cm${}^{3}$·m${}^{-2}$·h${}^{-1}$—diffusing from the human skin to the outer air. The phenomenon of CO${}_{2}$ exhalation—a.k.a. cutaneous respiration—through the skin was particularly studied in the early 20th century and Fitzgerald [335] wrote a complete and critical review on the topic in 1957. With the help of his work and others, the following information could be gathered:–The exhalation rate of CO${}_{2}$ through the skin increases with an increasing temperature.–The CO${}_{2}$ production rate of the tissues increases with temperature (due to an increase in tissues metabolism).–Skin humidity, or the presence of various chemicals can change the CO${}_{2}$ exhalation rate through the skin [336].

The increase in CO${}_{2}$ exhalation rate through the skin with an increasing temperature is widely accepted by the community [324,335,337] and is often cited as a reason for heating the skin whilst measuring transcutaneous CO${}_{2}$. Justifying this, the remarkable works of Shaw, Whitehouse et al. in the early 1930s [323,336,338] are of particular interest. Their main findings and that of others are presented in Table 2. Some values that can be found in Fitzgerald’s review [335] were omitted on purpose due to the non-availability of the publications from which they emanate, or because of unusable units.

As pointed out by Fitzgerald, the main drawback of these studies is the fact that the measured temperature is that of the air surrounding the skin, and not that of the skin itself—although the two correlate to some extent [342]. Further research on the influence of skin temperature on its CO${}_{2}$ exhalation rate would thus be welcome. A plausible explanation for this lack of knowledge is the clinical aspect of further studies led in the 1950s–1990s, aiming at developing a new sensorless intrusive than the more traditional blood gas analysis [4]. In this context, researchers focused less on the response time of the sensor and more on the clinical relevance of the measured tcpCO${}_{2}$ value [333,343,344].

Nevertheless, the existence of the influence of skin temperature on CO${}_{2}$ exhalation rate can reasonably be assumed to be true. It has been even further justified by the partially lipid structure of the skin, which strongly affects its permeability towards gases with changes in its temperature [324]. Further still, diminishing response times with an increase in skin temperature were also demonstrated for tcpCO${}_{2}$ sensors in a clinical setting [345,346].

#### 4.2.3. Impact on the Sensor Form Factor in a Closed-Chamber Design

Of the many constraints imposed by transcutaneous CO${}_{2}$ sensing, the limited CO${}_{2}$ exhalation rate by the skin is probably the most challenging one. Indeed, let us consider the simplified, generic, closed-chamber sensor model presented in Figure 14 (Left). In this crude model, the skin is considered as a membrane of thickness *e* and diffusivity *D* (m${}^{2}$·s${}^{-1}$) towards CO${}_{2}$. The sub-cutaneous tissues are considered as a semi-infinite medium wherein the pCO${}_{2}$ is constant and equal to tcpCO${}_{2}$. Finally, the sensor of total volume $V_{Se}$ (contact area with the skin $S_{Se}$ and height $h_{Se}$) of internal pressure $p_{Se}{\mathrm{CO}}_{2}$ is enclosed into a supposedly gas-tight body, with non-leaking contacts between the sensor body and the skin. Using Fick’s first law of diffusion applied to CO${}_{2}$ in the membrane with *x* the vertical axis leads to:(11)$$J_{S}=-D\cdot \frac{dC(x)}{dx}$$
wherein $J_{S}$ is the CO${}_{2}$ diffusion flux per unit area (mol·m${}^{-2}$·s${}^{-1}$) and C the CO${}_{2}$ concentration (mol·m${}^{-3}$). Supposing a constant total pressure and integrating over the membrane’s thickness under the hypothesis that $p_{Se}{\mathrm{CO}}_{2}$ is null at *t* = 0 yields—see Appendix A for complete details and calculations:(12)$$p_{Se}{\mathrm{CO}}_{2}\left(t\right)={\mathrm{tcpCO}}_{2}\cdot \left(1-exp\left(-\frac{t}{\tau }\right)\right),\;\mathrm{with}\;\tau =h_{Se}\cdot \frac{e}{D}$$

Which is a classic first-order system response, as described in Figure 14 (Right). The response time of the sensor used in this crude model is thus the only function of two elements: the dimensions of the sensor itself, given by $h_{Se}$—that is the $V_{Se}/S_{Se}$ ratio—on the one hand, and the properties of the skin membrane—the e/D ratio—on the other hand. Quite intuitively, a large surface to volume ratio of the sensor, a thin skin membrane, and an elevated diffusivity all lead to a faster sensor.

From this crude model, we can learn that in order to decrease the response time of a transcutaneous sensor one can either act on the sensor form factor, by designing a thinner sensor, or on the skin diffusivity and thickness. Indeed, increasing the skin diffusivity can be achieved by heating—hence the positive influence of skin temperature on CO${}_{2}$ diffusion rate discussed in Section 4.2.2—while stripping the skin upper layers reduces *e*—mentioned in Section 4.1. However, the last two options can be uncomfortable for the patient.

Estimating the e/D ratio is thus necessary to get an idea of $h_{Se}$ values compatible with a reasonable sensor response time τ. It can be demonstrated—see Appendix A—that:(13)$$\tau =h_{Se}\cdot \frac{{\mathrm{tcpCO}}_{2}}{Q_{S}\cdot p_{0}}{|}_{t=0}$$
with $Q_{S}$ the CO${}_{2}$ exhalation rate through the skin and $p_{0}$ the total ambient pressure. From Section 4.2.2, $Q_{S}\approx 100$ cm${}^{3}$·m${}^{-2}$·h${}^{-1}$ for normocapnic—tcpCO${}_{2}$ ≈ 5.3 kPa (40 mmHg)—subjects is deemed plausible. These crude estimations lead to the response time presented in Figure 15. As can be seen, for a sensor to have a reasonable response time—e.g., below 10 min—it must be relatively thin, in the range of 100 µm or below. Given the approximations made above, and the hypotheses presented in the Appendix A, the reader should bear in mind that this 100 µm threshold is given as an order of magnitude of targeted sensor height.

### 4.3. Recommendations for a Closed-Chamber Design

Summarising the above, when designing a tcpCO${}_{2}$ sensor two main parameters should be carefully taken into account: (i) the skin temperature—as it influences both the sensor *response time* through the CO${}_{2}$ rate of diffusion and its *accuracy* through the paCO${}_{2}$/tcpCO${}_{2}$ correlation—and (ii) the form factor of the sensor itself—as a crude analysis of a simplified sensor revealed that a volume to surface ratio of approximately 100 µm or below is required to yield reasonable response times. Considering a transcutaneous application, the CO${}_{2}$ measuring techniques presented in Section 2 are reviewed below.

At first, only some of them can deal with a height limitation of only 100 µm. In that respect, NDIR, photoacoustic, time-of-flight, and acoustic attenuation sensors cannot be miniaturized to reach such a thickness. Second, the use of metal oxide thin films as well as electrochemical cells should be highly discouraged since both operate at several hundreds of degrees (°C). As such, using them against the skin may entail a significant safety hazard. Then, conductometric and ISFET-based sensors as well as the Stow-Severinghaus electrode all suffer from important drift issues, making them unusable for more than a few hours, or days. Optical fiber-based sensors, for their part, require an expensive and bulky measurement apparatus that seems hardly embeddable into a wearable device.

Remain the ionic liquid, graphene, and dye-based sensors. While recent research on ionic liquid-based sensors showed definite progress, they are still extremely sensitive to O${}_{2}$ and often exhibit particularly elevated recovery times [226,232]. Similarly, recent developments pave the way for graphene-based CO${}_{2}$ sensors [196], but their important cross-sensitivity towards humidity precludes their use without an additional humidity sensor for referencing, due to the skin humidity. Thus, the option of choice appears to be the more mature dry dye-based sensors for they can yield thin enough, fast-responding, and long lasting pCO${}_{2}$ sensors—see Section 2.3.4. In addition, dye-based sensors can be dissociated into a CO${}_{2}$ sensing part, on the one hand, and an optical probing part on the other hand, as presented in Figure 16. This design is especially interesting since it makes it possible to use a disposable sensing patch that can be replaced every few days while the sensing head can be reused indefinitely. Finally, the multilayer structure of the patch seems well adapted to existing manufacturing processes—roll-to-roll in particular—for mass production [347,348].

### 4.4. Other Perspectives

Apart from a closed-chamber design—for which dye-based sensing seems to be the most appropriate sensing technique—transcutaneous CO${}_{2}$ monitoring may also be envisioned using a rate-based approach, or Diffuse Reflectance Spectrometry (DRS).

#### 4.4.1. Rate-Based Approach

The rate-based approach has already been presented in Section 4.1, and corresponding theoretical developments can be found in the work of Chatterjee et al. [318]. The main advantage of this approach is that it releases the form factor constraint highlighted above in the closed-chamber case. Thus, NDIR sensing can be used instead of a dye-based thin film, with better lifetime, accuracy, and selectivity over the latter. These advantages made NDIR sensing the foremost choice of the authors who considered using a rate-based approach for transcutaneous CO${}_{2}$ sensing [317,318,319,321,322].

The biggest issue with this approach is the inter-subject variability in skin diffusivity towards CO${}_{2}$. Indeed, all studies reporting measurements of the skin CO${}_{2}$ exhalation rate mention this variability, with a factor 1.5–2 being not unheard of between the exhalation rates of two individuals, even at the same skin site and in the same temperature conditions (see references in Table 2). This variation among individuals precludes the measurement of *absolute* tcpCO${}_{2}$ by mean of a rate-based approach without further improvements of the technique, as demonstrated by several authors who could only report trend analyses of skin CO${}_{2}$ exhalation rate, and no rate-based absolute tcpCO${}_{2}$ readings [319,321,322]. Further still, the recent work of Iitani et al. [320] clearly illustrates the afore-mentioned inter-subject variability with a factor of more than ten between the exhalation rates of two subjects (S2 and S6 on their Figure 4-H). There may be an exception concerning neonates, however, whose exceedingly thin skin may be more homogeneous than that of adults, leading to less variability in their skin CO${}_{2}$ exhalation rate, and allowing Chatterjee et al. [317] to demonstrate a significant correlation between skin CO${}_{2}$ exhalation rate and tcpCO${}_{2}$.

To overcome this challenge, several tracks can be followed. At first, further studies on the CO${}_{2}$ exhalation rate through human skin would be welcome, since the currently available literature is aging, and rarely include more than a handful of subjects. In addition, the temperature dependence of the exhalation rate also needs to be carefully studied, by regulating the skin temperature, and not that of the ambient air surrounding it. This would greatly help to better characterize the intra- and inter-subject variations in CO${}_{2}$ exhalation rate with skin site, skin temperature, or time of the day.

Second, dual membrane techniques—as that presented by Targett et al. [330]—may be adapted to modern sensing techniques in a setup and compactness close to that of Grangeat et al. [321] prototype, using two different sensing channels, for instance.

Third, the need for an absolute tcpCO${}_{2}$ reading could also be mitigated in two ways. First off, by mean of a punctual per-subject calibration, following the example of non-invasive, cuffless, photoplethysmography (PPG)-based blood pressure measurements [349,350]. In this sensing scheme, the actual tcpCO${}_{2}$ could be measured from time to time using a reference tcpCO${}_{2}$ monitor, whose measurements would serve to calibrate a lighter, wearable, rate-based device. This would allow sharing the main tcpCO${}_{2}$ monitor among multiple patients, in a clinical setting. Alternatively, it could also be envisioned that rate-based devices do not provide any absolute tcpCO${}_{2}$ reading. Rather, they may provide the user with trend analysis only, raising alarms following the crossing of certain thresholds or abrupt variations. This last approach, however, may be difficult to introduce into medical practice without further clinical studies, even though the clinical interest of similar variability metrics for other physiological variables—such as heart rate [351] or oxygen saturation [352,353]—is encouraging in this perspective.

#### 4.4.2. Diffuse Reflectance Spectometry (DRS)

As stated in Section 2.2.1, dissolved CO${}_{2}$ measurements can be performed not only in the gaseous phase, but also in aqueous solution [47]. This result was also demonstrated for whole blood samples, the pCO${}_{2}$ of which could be retrieved with a standard error of about 5 mmHg, using infrared spectroscopy in the 1024–2290 nm range [354]. Further still, it has been recently suggested in a patent that transcutaneous CO${}_{2}$ measurements could be performed using mid-infrared LEDs of wavelength “between about 2 µm and about 18 µm” [355].

Although patents do not constitute scientific evidence [356], the idea has been put forth, and further research would be interesting on the topic, in order to assess the feasibility of such transcutaneous CO${}_{2}$ measurements. *Prima facie*, two major challenges conceivably need to be solved for DRS-based transcutaneous measurements.

First, contrary to pulse oximetry, the possibility to measure a pulsatile absorbance component due to CO${}_{2}$ dissolved in the arterial blood by transcutaneous DRS probing in the mid-infrared is vastly unknown. If possible though, the whole theoretical background developed over the years for pulse oximetry and PPG signal processing [357] could be translated to CO${}_{2}$ measurements.

Second, the penetration depth of mid-infrared light in the human tissues may also be a source of concern, especially because of the elevated infrared absorbance of water. Indeed, water absorbance becomes predominant in the tissues in the near and mid infrared [358], with values over 1 cm${}^{-1}$ above 1150 nm, and over 5 cm${}^{-1}$ above 1380 nm [359]. This may cause serious difficulties for mid-infrared tissue probing and lead to a very low signal-to-noise ratio, which may be the reason why Marasco et al. suggest the use of a lock-in amplifier for transcutaneous CO${}_{2}$ measurement in their patent [355].

In both cases, heating the skin to provoke local hyperaemia—as mentioned in Section 4.2.1—may tackle the above-mentioned probing depth issue, by bringing the CO${}_{2}$ content of subcutaneous tissues close to that of arterial blood.

### 4.5. Synthesis

tcpCO${}_{2}$ is the least invasive and the most accessible proxy for paCO${}_{2}$—which is of major clinical significance—given that the skin temperature is at least above 37 °C (Section 4.2.1). tcpCO${}_{2}$ may be measured by two means: either exploiting the cutaneous respiration phenomenon, or by direct infrared tissue probing, using DRS.

While the feasibility of the latter technique has yet to be demonstrated for in vivo transcutaneous CO${}_{2}$ monitoring, it seems like an appetizing proposal (Section 4.4.2). Indeed, it would require merely a mid-infrared source and detector—which are readily available components, see Section 2.2.1—and conceivably the same kind of electronic as that already embedded in commercially available pulse oximeters. Yet, such an application of DRS remains vastly hypothetical at the time of writing this review, and further research would be needed on the topic, so as to consolidate the above-mentioned research works.

Cutaneous respiration, on its part, has long proven to be a reliable access route to tcpCO${}_{2}$, by capturing the CO${}_{2}$ exhaled by the skin inside the embodiment of a CO${}_{2}$ sensor (Section 2.3.2). A simple model for such a closed-chamber sensor design has been extensively described, as well as its expected response time as a function of its form factor (Section 4.2.3). We conclude that in such a closed-chamber design, the compactness of the CO${}_{2}$ sensor is crucial in order to reach reasonable response times. Taking into account other merit criteria such as response time, long-term stability, and accuracy, we put forth the application of dye-based CO${}_{2}$ sensing to serve such a purpose.

Alternatively, a rate-based approach may be envisioned using a constant flow of air or N${}_{2}$ passing over the skin, and being analyzed thereafter (Section 4.4.1). The main advantage of this technique is that the form factor of the sensor itself is not as much a source of concern as in the closed-chamber setup. Thus, other sensing techniques can be envisioned such as NDIR CO${}_{2}$ sensing. However, several challenges remain, precluding rate-based approaches to yield an absolute tcpCO${}_{2}$ measurement. In particular, a better characterization of human skin CO${}_{2}$ exhalation rate, dual membrane approaches, or self-calibration from time to time are all interesting leads that need further exploration.

## 5. Conclusions

Building on previous knowledge, we present in Section 2 an extensive review of CO${}_{2}$ sensing techniques, the main characteristics of which are summed up in Table 1. Then, we emphasize the clinical importance of CO${}_{2}$ monitoring in humans in Section 3. The latter monitoring can be performed through three distinct modalities: (arterial) blood puncture, airway capnometry, or transcutaneous capnography. In the first and last cases, miniature Stow-Severinghaus electrode or ISFET are used for pCO${}_{2}$ sensing, while NDIR sensors are used for airway capnometry.

Among these three modalities, transcutaneous monitoring is the least invasive one, while still providing a clear picture of a patient’s haemodynamic status. However, current commercially available sensors based on the Stow-Severinghaus electrode suffer from several drawbacks—mainly an important drift and an elevated cost—and an alternative would be highly desirable.

In this view, we first present several works aiming at the same target and discuss their advantages and drawbacks. We then propose a simple model of transcutaneous sensor, taking into account early studies on human cutaneous respiration in order to emphasize the importance of the sensor form factor in transcutaneous sensing. More specifically, we conclude that a volume to surface ratio—or equivalent height—as low as possible—ideally below approximately 100 µm—must be achieved in order to obtain reasonable, clinically exploitable response times using a closed-chamber sensor design. The CO${}_{2}$ sensing techniques presented in Section 2 are then reviewed against these requirements in terms of form factor, cost, and performance. Under the current state of the art, we conclude that a dry dye-based sensor in the form of a thin-film patch seems the most promising approach to develop a new kind of closed-chamber tcpCO${}_{2}$ sensor.

Finally, perspectives for alternative transcutaneous probing modalities are also discussed, with an emphasis on two exploratory paths that may be pursued in the near future: DRS, and the rate-based approach. At the time of writing this review, these two alternatives seem both challenging, bringing their share of exciting research questions that have yet to be answered.

## Figures and Tables

**Figure 1 sensors-22-00188-f001:**
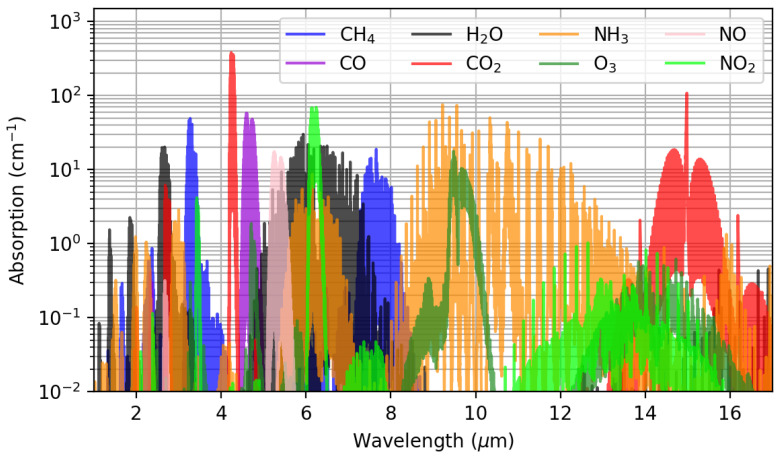
Mid infra-red absorption spectra of various gases, only CO${}_{2}$ absorbs at 4.26 µm. Data source: HITRAN database [28]. A Lorentzian broadening profile was considered for a dilution in air at 1 atm and 296 K.

**Figure 2 sensors-22-00188-f002:**
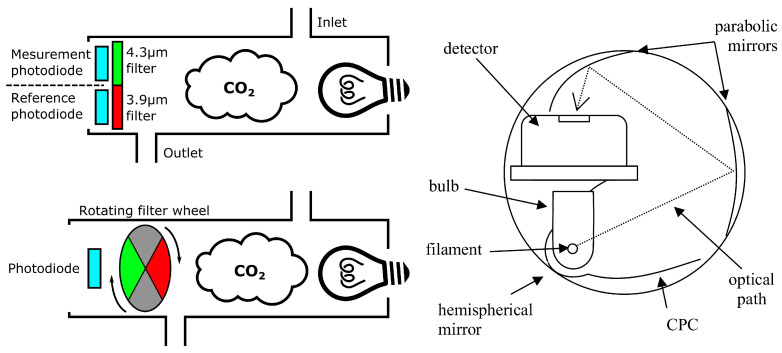
(**Left**) outline schematic of a space-interleaved (**top**) and time-interleaved (**bottom**) NDIR CO${}_{2}$ sensor. In the time-interleaved sensor, a rotating filter wheel acts as a chopper, with two opaque sectors, and two sectors equipped with 4.3 µm and 3.9 µm bandpass filters, respectively. (**Right**) a more complex design allowing for longer light paths in a compact device. The detector has two channels, a measurement and a reference one, as in the upper left scheme. CPC stands for Compound Parabolic Collector, a type of light concentrator. Reproduced with permission from Hodgkinson et al. [30].

**Figure 3 sensors-22-00188-f003:**
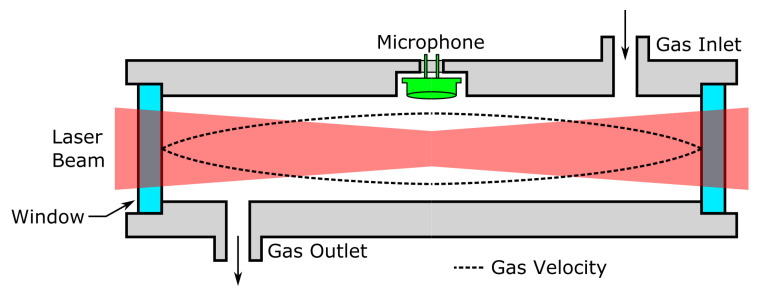
Outline schematic of an organ-pipe-like resonant acoustic cell. The cell consists of a pipe closed at its two ends with optical windows. A laser beam at 4.26 µm is then pulsed at the resonant frequency of the pipe, which forms a λ/2 resonator. For instance, if a 32,768 Hz quartz tuning fork is used, λ≈10 mm and a pipe length of ∼5 mm would be ideal. A velocity-sensitive microphone may be placed at half the pipe as depicted. Alternatively, a pressure-sensitive microphone would rather be placed near one of its ends.

**Figure 4 sensors-22-00188-f004:**
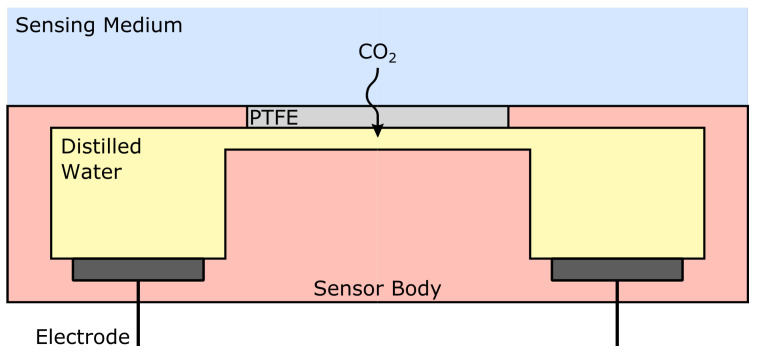
Outline of a static conductometric cell. CO${}_{2}$ diffuses into distilled water, modifying the concentration of H${}_{3}$O${}^{+}$, HO${}^{-}$ and HCO${}_{3}^{-}$ and thus the conductivity of the solution, which may be measured using alternative current to avoid polarisation by mean of two electrodes.

**Figure 5 sensors-22-00188-f005:**
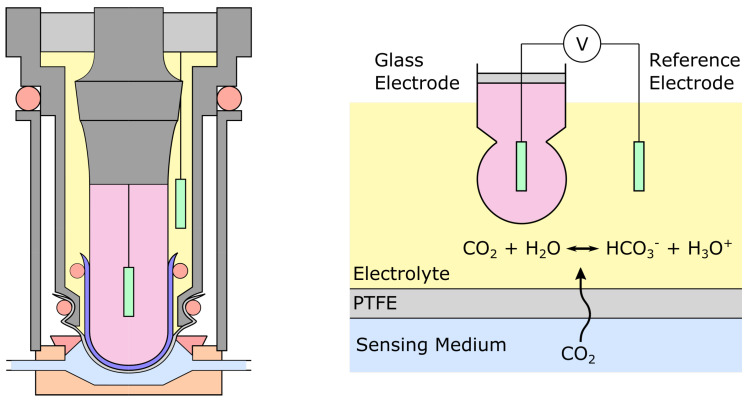
(**Left**) the Severinghaus electrode, an improvement of the Stow electrode, as first described in the 1958 publication [67]. The colour of the different elements matches that of the simplified schematic on the right. (**Right**) outline schematic of the Severinghaus electrode. CO${}_{2}$ diffuses through a PTFE membrane into an electrolyte, causing a change in the pH of the latter. Such a change is then recorded by the mean of a pH-meter that consists of a glass electrode and a reference electrode.

**Figure 6 sensors-22-00188-f006:**
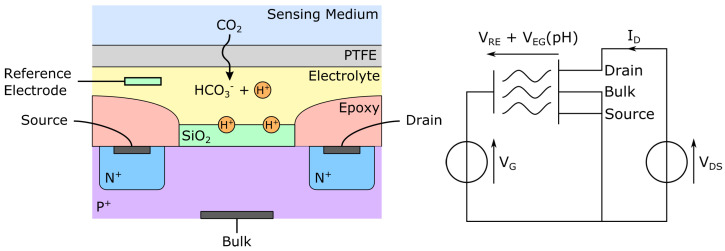
(**Left**) outline schematic of an ISFET CO${}_{2}$ sensor. CO${}_{2}$ diffuses from the analyte inside the inner electrolyte through a PTFE membrane, changing the pH of the electrolyte and generating hydronium ions (abbreviated H${}^{+}$). The ions will penetrate the upper layer of the porous SiO${}_{2}$ gate insulator, influencing the P${}^{+}$ substrate underneath, and thus the conductance of the transistor. (**Right**) basic electrical implementation schematic of an ISFET sensor. The actual grid-source potential seen by the transistor is the sum of the applied grid voltage $V_{G}$, the voltage between the reference electrode and the electrolyte $V_{RE}$, and that between the electrolyte and the gate insulator $V_{EG}$, itself function of the pH of the electrolyte.

**Figure 7 sensors-22-00188-f007:**
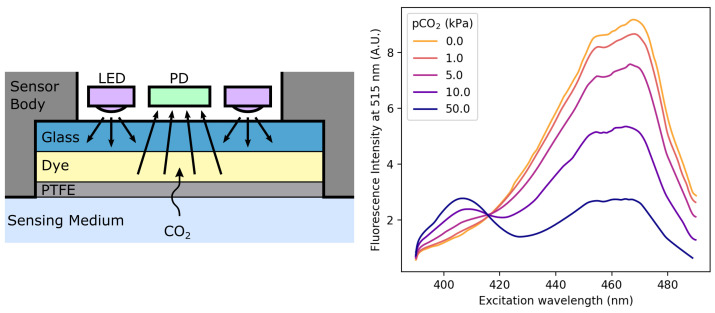
(**Left**) outline schematic of a typical dye-based CO${}_{2}$ sensor. The LEDs illuminate a pH-sensitive dye, either dissolved in an aqueous solution or a polymer, through a layer of transparent substrate—here a glass slide. The dye is protected from ionic contamination by a PTFE, gas-permeable, ion-impermeable membrane. The LEDs and photodiode may additionally be covered by excitation or emission filters in case of fluorescence measurement. (**Right**) the fluorescence excitation spectra of 1 mM HPTS in carbonated distilled water equilibrated with different pCO${}_{2}$ values. Data source: Uttamlal et al. [90].

**Figure 8 sensors-22-00188-f008:**
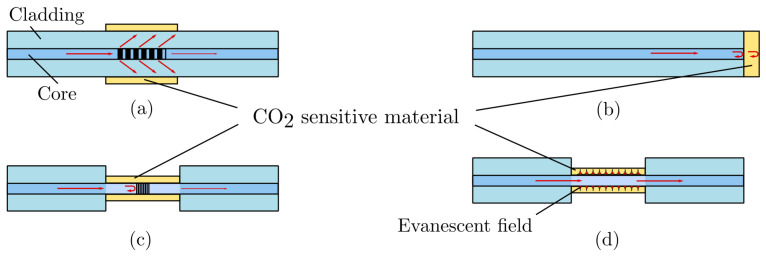
“Various techniques that have been employed in CO${}_{2}$ sensors to cause light propagating in the core to interact with the surrounding environment: (**a**) [Long Period Grating]; (**b**) extrinsic Fabry-Perot cavity; (**c**) [Fiber Bragg Grating] and; (**d**) etched cladding. All the examples here use a material which demonstrates CO${}_{2}$ sensitivity and the subsequent change in coating upon exposure to CO${}_{2}$ is sensed by the interacting light”—Text and figure reproduced with permission from Barrington [20].

**Figure 9 sensors-22-00188-f009:**
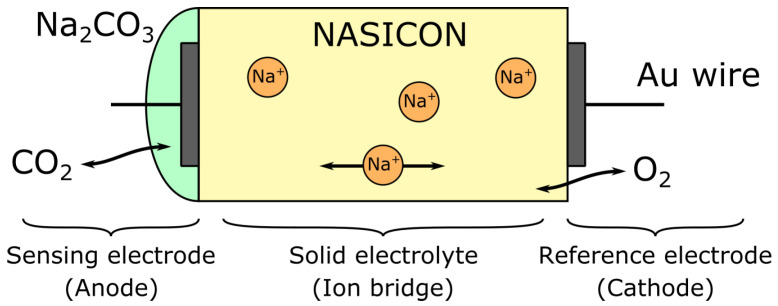
Outline schematic of a potentiometric CO${}_{2}$ sensor as described by Maruyama et al. [210]. The NASICON sandwich with one end covered in sodium carbonate makes an electrochemical cell, the electromotive force of which yields pCO${}_{2}$.

**Figure 10 sensors-22-00188-f010:**
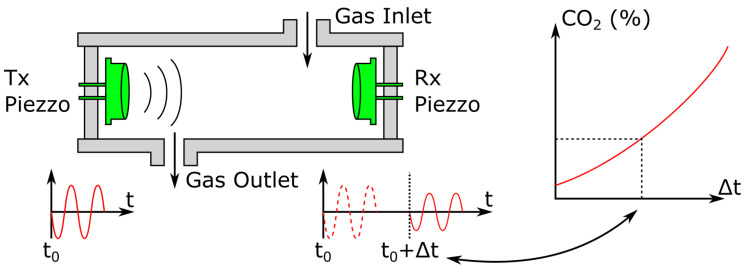
Outline schematic of a time of flight CO${}_{2}$ sensor as described by Joos et al. [245]. Two ultrasound (40 kHz) transducers—emitter (Tx) and receiver (Rx)—are placed at both ends of an acoustic chamber which may consist in a simple tube. A burst of ultrasound is emitted at $t_{0}$ (on the **left**) and arrives at $t_{0}+\Delta t$ at the receiver (on the **right**). The Δt value can then yield the percentage of CO${}_{2}$ in the gas mixture using velocity-based calculations.

**Figure 11 sensors-22-00188-f011:**
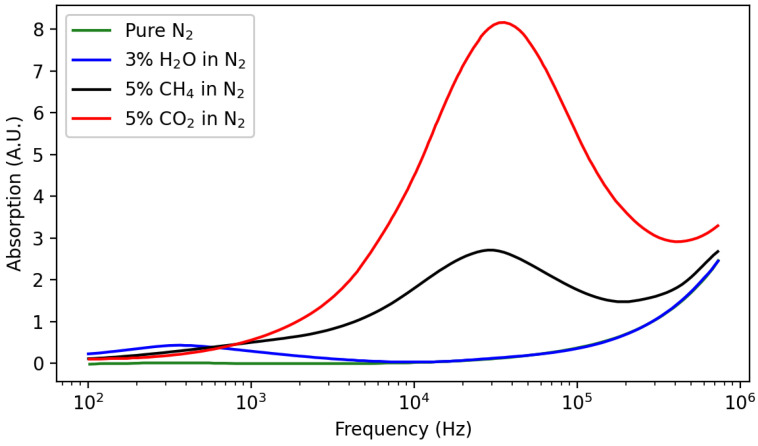
Acoustical absorption spectra of diverse nitrogen mixtures. Data source: Petculescu et al. [250].

**Figure 12 sensors-22-00188-f012:**
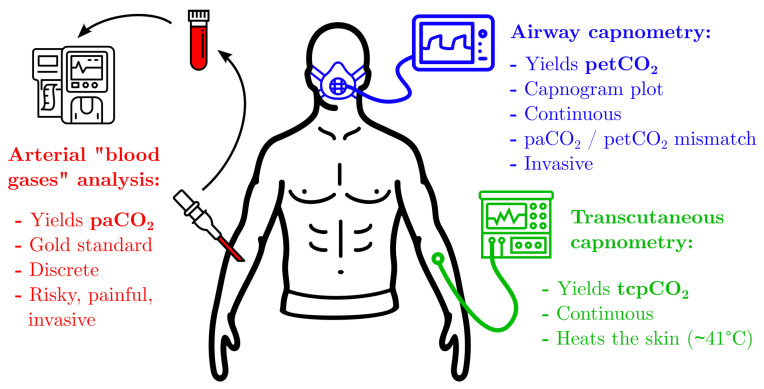
The three in vivo CO${}_{2}$ probing modalities and their key features.

**Figure 13 sensors-22-00188-f013:**
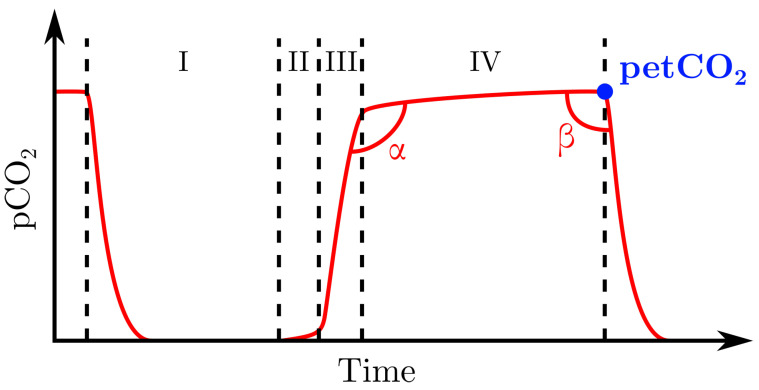
A capnogram yields important clinical clues on the state of the patient or quality of their intubation by analysing the different breathing phases—inspiration (I), and expiration: dead-space volume (II), mixed dead-space and alveolar air (III) and alveolar air (IV)—and the two α and β angles [292,293]. More narrowly, capnometry is only interested in knowing the more concise end-tidal pCO${}_{2}$ value reached at the end of the plateau (IV): petCO${}_{2}$.

**Figure 14 sensors-22-00188-f014:**
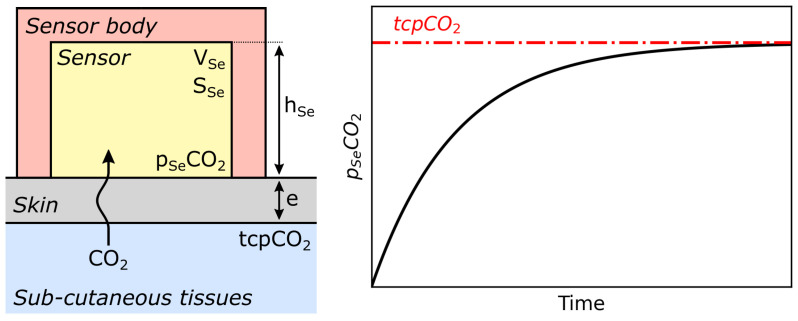
(**Left**) crude closed-chamber sensor model. Please note the colour matching between this schematic and previous Figures—e.g., Figure 4, Figure 5, Figure 6 and Figure 7—blue for the analyte medium, yellow for the sensor itself. (**Right**) typical evolution of $p_{Se}$CO_2_ against time when applying the sensor against the skin.

**Figure 15 sensors-22-00188-f015:**
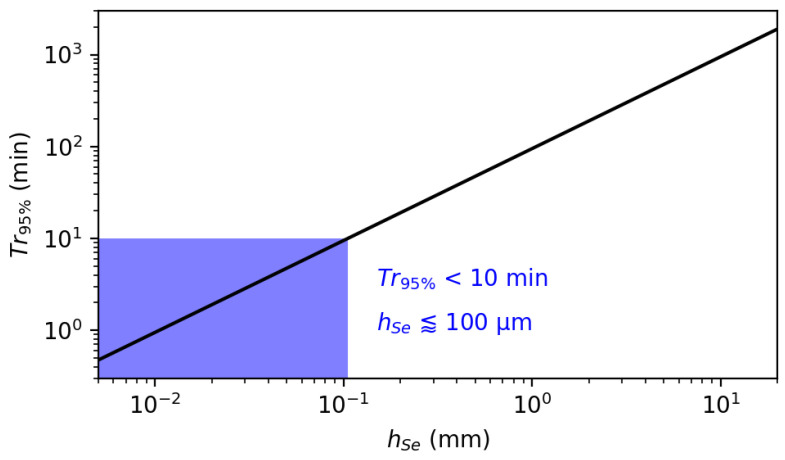
95% response time ($Tr_{95\%}\approx 3\cdot \tau$) of a closed-chamber sensor of height $h_{Se}$. The blue area underlines the portion of the line with a response time below 10 min, which corresponds to a sensor height below approximately 100 µm.

**Figure 16 sensors-22-00188-f016:**
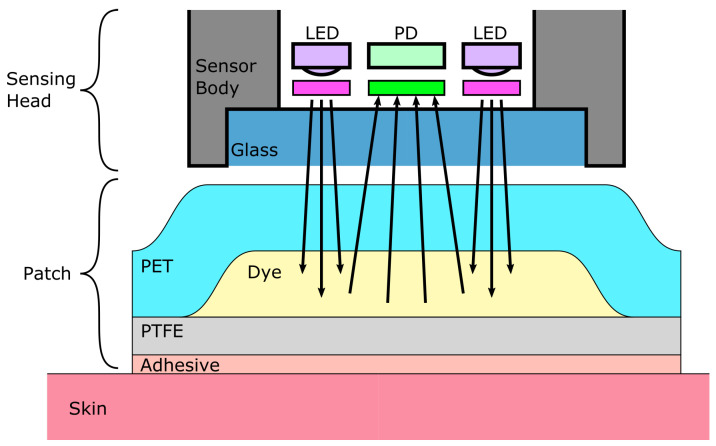
The dye-based fluorescent transcutaneous sensor consists of two parts. (**Top**) a sensing head with the LEDs and photodiode covered with their emission and reception filters (bright purple and green rectangles) and enclosed in a solid body, protected by a covering glass. (**Bottom**) a multilayer sensing patch stuck on the skin, consisting in—from top to bottom—a transparent layer, impermeable towards CO${}_{2}$—e.g., PET—a layer of CO${}_{2}$ sensitive dye in a polymer matrix, a layer of ion-impermeable, CO${}_{2}$-permeable material such as PTFE for CO${}_{2}$ diffusion from the skin into the dye, and a thin adhesive layer to maintain the patch onto the skin. See Section 2.3.4 for a detailed review of material choices as well as their advantages and drawback, when designing a dye-based CO${}_{2}$ sensor.

**Table 1 sensors-22-00188-t001:** An overview of the presented techniques. (*∞*): expected to be at least several years. ${}^{\left(a\right)}$: limited primarily by the diffusion of the analyte gas inside the enclosure of the sensor. Please refer to detailed explanations in Section 2.6.

Criterion → ↓ Technique	Lifetime	Calibration Need	Humidity Cross-Sensitivity	Accuracy	Response Time	Miniaturisation Potential	References
NDIR (Section 2.2.1)	≥15 years	No	No	0.03%	≤1 s ${}^{\left(a\right)}$	Moderate (down to 10×10×0.5 mm${}^{3}$)	[32,33,34,36]
Photoacous. (Section 2.2.2)	? (*∞*)	No	No	1%	≤1 s ${}^{\left(a\right)}$	Low (∼cm)	[51,257,258]
Wet Conduct. (Section 2.3.1)	?	Yes (0.4% drift/h)	Yes (dries out)	?	5 s	High (down to 100 µm thickness)	[60,62]
S-S elec. (Section 2.3.2)	80 days	Yes (0.03 mV/h)	Yes (dries out)	0.03% (0–100 mmHg)	30 s	High (down to 10 µm thickness)	[67,69,73,259]
ISFET (Section 2.3.3)	≥30 days	Yes (0.23 mV/h)	Yes	?	60 s	High (down to 100 µm thickness)	[79,88]
Dye-based (Section 2.3.4)	19 months	No	Yes	0.1%	≤1 s	Excellent (down to 1 µm thickness)	[56,103,112,122,123,260]
Optical fiber (Section 2.3.5)	?	?	Yes	4.1%	40–75 s	Excellent (down to 55 nm coating thickness)	[145,146,147]
Metal Oxide Ads. (Section 2.4.1)	≥5 months	No	Yes (low)	1.6%	30 s	Excellent (down to 125 nm thickness)	[155,163,164,171,175,189,261]
Graphene (Section 2.4.2)	≥90 days	No	Yes	0.8%	3 s	Excellent (down to 3 nm thickness)	[194,196,207]
Electro. Cells (Section 2.4.3)	≥2 years	No	No	4%	2–4 s	Excellent (down to 1 µm thickness)	[213,214,221,262]
Ionic liquids (Section 2.4.4)	≥4 months	No	No	2.05%	40 s	Excellent (down to 430 nm thickness)	[226,231,233,234,238]
Time of flight (Section 2.5.1)	? (*∞*)	No	Yes (low)	0.3% (0–10% range)	≤1 s ${}^{\left(a\right)}$	Low (∼cm)	[245]
Acous. Att. (Section 2.5.2)	? (*∞*)	No	?	?	≤1 s ${}^{\left(a\right)}$	Low (∼cm)	[250,254]

**Table 2 sensors-22-00188-t002:** CO${}_{2}$ exhalation rate through the skin in the literature.

Exhalation Rate [cm${}^{3}$·m${}^{-2}$·h${}^{-1}$]	Temp. [°C]	Num. of Subjects	Ref.
25–120	22–36 (air)	2	Shaw (1929) [336]
10–160	25–37 (air)	1	Shaw (1930) [323]
40–125	25–30 (air)	2	Ernstene (1932) [327]
12–143	23–37 (air)	1	Whitehouse (1932) [338]
180–2500 ^†^	-	1	Adamczyk (1966) [325]
25–87	-	5	Thiele (1972) [339]
11–28	25–35 (air)	3	Frame (1972) [340]
50	-	27	Levshankov (1983) [326]
131–206	36 (skin)	14	Eöry (1984) [341]

^†^ Axilla measured value, possibly erroneous.

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
