# Peer review of "Carbon Dioxide Sensing—Biomedical Applications to Human Subjects"

_sensors, 2021, doi:10.3390/s22010188_

Round 1

Reviewer 1 Report

This paper gives a very interesting overview on new technologies for carbon dioxide sensing and their applications for transcutaneous measurement. I am very impressed by the number of papers and the number of technologies that have been covered. It is hard for me to review all of them.

Some technologies are not covered, such as the reflectance measurement of the skin carbon dioxide content using IR light illuminating the skin and reflected by the skin as for oximetry DRS measurement [1], the use of temperature modulation associated with MOX sensor measurements [2] [3] [4], or graphene carbon dioxide sensors [5] . A special section dedicated to membrane sensors might be also considered [6]. The membrane is a key component for some sensing technologies.

The analysis of the NDIR measurement technology could be improved. The measurement principle is based on the Beer Lambert law. But in general, there is large variety of propagation path between the emitter and the receiver. Thus the relationship between the measured concentration and the logratio of the dual wavelength measurement is no more linear. This means also that is has to be calibrated. Also, when considering transcutaneous measurement, the humidity level is very high due to sweat effect, and much higher that the carbon dioxide concentration. Thus, the effect of humidity has to be corrected. See the Grangeat et al papers you are referencing for more details [7] [8] [9], or the following article. The section 2.2.1 has to be corrected and enhanced.

But for me the week point of this paper is on the fluidic analysis. Only a simple measurement principle is described in this paper associated with the static closed chamber geometry. In this geometry, you need to wait for an equilibrium between the carbon dioxide pressure in the blood and the carbon dioxide pressure in the measurement chamber and this equilibrium is reached in a passive way. Thus, the time to reach this equilibrium is very long. This is why in new studies such as the one described by Chatterjee et al or Grangeat et al you are referencing, a dynamic flow through chamber methodology is considered resulting in a much faster acquisition principle. Then, the conclusion you are drawing comparing all the technologies might be different since in the geometry you are proposing you are still working on a passive diffusion through the skin, which is the limiting factor. In the solution you are proposing, there is no circulation of the carbon dioxide and thus the response time to follow the dynamic variation of the carbon dioxide blood content might still remain long with respect to the physiological variations.

For the methodology first presented by Chatterjee et al. several following papers have been published which should be also referenced [10], [11], [12].

Hereafter are some suggested corrections:

  • Line 160 suppress one “of”
  • Figure 8: the explanations are not clear. Some texts are within []: why?
  • Line 382: you give an order of magnitude for the time response of a MOX sensors of up ta several hours. But in the most often case the time response is much faster, in the order of 1 second or less
  • Section 2.4.1 on MOX sensor should be completed to include temperature modulation.
  • Line 480: this is not clear (more often than not probed)
  • Table 1: it should be clear that this is the response time of the sensing component but not of the sensing device which should include at least the carbon dioxide transport time response
  • Table 1: the section on calibration need is not clear. I don’t know any sensor that do not need to be calibrated, at least once.
  • The section 3.3.1 should be completed. The carbon dioxide is diluted in several species, either molecules or ions. The partial pressure is characterizing only the molecular species. Also, some of the carbon dioxide is transported by the red cells. Thus, the partial pressure is only an indicator of the total carbon dioxide diluted.
  • Table 2 is not exhaustive
  • Section 4.2.2 : the diffusion rate depends from the temperature. But also the carbon dioxide solubility.
  • Line 805: Js is a diffusion flux per unit area
  • Line 816: heating the skin increases the diffusion flux but decreases the solubility. Also the exchange surface is reduced by the heating circuits.
  • 15: This should be make clear that this is the response time in static closed chamber mode
  • Section 4.3 should be moderated to explain that these recommandations concern the static closed chamber mode. Or you need to examine more precisely the requirements to work in a dynamic mode.
  • Lines 866 to 868: This is your conclusion for the static closed chamber mode.
  • Line 886: “mentioned” to be corrected

For supplementary materials:

  • Title: transcutaneous
  • Section 2: it should be made clear that you are working in a closed chamber mode and your reservoir is closed.
  • Section 3: Js is a diffusion flux per unit area
  • Section 3: “capacitive effect of the skin is negligible” is not a correct assumption. The carbon dioxide solubility within the stratum corneum is nearly 3 times higher than the one in the blood. In the model, you should include the solubility or Henry coefficients of the blood and of the skin.
  • The assumption that R . T . Kskin;CO2 factor in the 0.6 -1.0 range is not in agreement with the assumption proposed by Scheuplein and Blank [13] for who the Henry constant of the stratum corneum is nearly 1.6.
  • The influence of blood is not included in the model. Blood has an influence on the carbon dioxide diffusion which cannot be neglected. The blood flow has also an influence on the carbon dioxide flow rate.

References:

[1]        Ch. Marasco et al, Metabolic rate measurement apparatus and method thereof, USA patent US20200113516A1.

[2]        R. Gosangi et R. Gutierrez-Osuna, « Active temperature modulation of metal-oxide sensors for quantitative analysis of gas mixtures », Sensors and Actuators B: Chemical, vol. 185, p. 201‑210, août 2013, doi: 10.1016/j.snb.2013.04.056.

[3]        A. Sudarmaji et A. Kitagawa, « Application of Temperature Modulation-SDP on MOS Gas Sensors: Capturing Soil Gaseous Profile for Discrimination of Soil under Different Nutrient Addition », Journal of Sensors, vol. 2016, p. 1‑11, 2016, doi: 10.1155/2016/1035902.

[4]        S. Madrolle, P. Grangeat, et C. Jutten, « A Linear-Quadratic Model for the Quantification of a Mixture of Two Diluted Gases with a Single Metal Oxide Sensor », Sensors, vol. 18, no 6, p. 1785, juin 2018, doi: 10.3390/s18061785.

[5]        F. Akhter, Md. E. E. Alahi, H. R. Siddiquei, C. P. Gooneratne, et S. C. Mukhopadhyay, « Graphene Oxide (GO) Coated Impedimetric Gas Sensor for Selective Detection of Carbon Dioxide (CO 2 ) With Temperature and Humidity Compensation », IEEE Sensors J., vol. 21, no 4, p. 4241‑4249, févr. 2021, doi: 10.1109/JSEN.2020.3035795.

[6]        T. Li, Y. Wu, J. Huang, et S. Zhang, « Gas sensors based on membrane diffusion for environmental monitoring », Sensors and Actuators B: Chemical, vol. 243, no Supplement C, p. 566‑578, mai 2017, doi: 10.1016/j.snb.2016.12.026.

[7]        P. Grangeat, S. Gharbi, M. Accensi, et H. Grateau, « First Evaluation of a Transcutaneous Carbon Dioxide Monitoring Wristband Device during a Cardiopulmonary Exercise Test* », in 2019 41st Annual International Conference of the IEEE Engineering in Medicine and Biology Society (EMBC), juill. 2019, p. 3352‑3355. doi: 10.1109/EMBC.2019.8857020.

[8]        P. Grangeat et al., « Evaluation in Healthy Subjects of a Transcutaneous Carbon Dioxide Monitoring Wristband during Hypo and Hypercapnia Conditions », in 2020 42nd Annual International Conference of the IEEE Engineering in Medicine & Biology Society (EMBC), Montreal, QC, Canada, juill. 2020, p. 4640‑4643. doi: 10.1109/EMBC44109.2020.9175876.

[9]        P. Grangeat, S. Gharbi, M. Accensi, et H. Grateau, « Etude d’un modèle linéaire quadratique appliqué à la mesure optique infrarouge du gaz carbonique sanguin émis par la peau », XXVIIème Colloque GRETSI, 26-29 août 2019, Lille, France.

[10]      M. Chatterjee et al., « A rate-based transcutaneous CO 2 sensor for noninvasive respiration monitoring », Physiol. Meas., vol. 36, no 5, p. 883‑894, mai 2015, doi: 10.1088/0967-3334/36/5/883.

[11]      M. Chatterjee, X. Ge, Y. Kostov, L. Tolosa, et G. Rao, « A novel approach toward noninvasive monitoring of transcutaneous CO2 », Medical Engineering & Physics, vol. 36, no 1, p. 136‑139, janv. 2014, doi: 10.1016/j.medengphy.2013.07.001.

[12]      X. Ge et al., « Development and characterization of a point-of care rate-based transcutaneous respiratory status monitor », Medical Engineering & Physics, vol. 56, p. 36‑41, juin 2018, doi: 10.1016/j.medengphy.2018.03.009.

[13]      R. J. Scheuplein et I. H. Blank, « Permeability of the skin », Physiological reviews, vol. 51, N°4, p. 702 - 747, 1971.

Reviewer 2 Report

The review paper about Carbon Dioxide Sensing is well written and a lot of different sensing principles with their pros and cros are presented.

The figures are clear and have enough details to understand them.

From my site, I have no points which a critical enough to do a second review.

Reviewer 3 Report

Dear colleagues!

The review concerns various aspects of carbon dioxide sensing and its biomedical applications to human subjects. The authors comprehensively described CO2 sensing techniques that are available at the moment, the main goals and constraints intrinsic to biomedical monitoring, and trends in transcutaneous sensing. The main points are interesting, the manuscript is concisely written and well illustrated. 

However, in my opinion, the authors should take a more critical approach to presentation and discussion of the materials. To be more precisely, the authors should give more stressing to advantages, disadvantages and prospects.

Round 2

Reviewer 1 Report

The paper has been improved and some alternative solutions have also been identified.

I am still uncomfortable with some points.

You criticize the rate based appraoch. But your computation is based on a rate assumption. 

You say you work with a rough approximated model. But you conclude your computation by giving the height the sensor should have with 3 digits.

About your conclusion, since the paper relies on data published in the litterature but not on experimental data, you should be carefull.

I am concerned by the fact the first author belongs to a private company and the study has been financed by this company. I would like to avoid this paper to be used for commercial purpose. It should be expressed clearly that for the discussion section, it reflects the point of view of the authors.

About the NDIR measurement and the effect of humidity, you give the value only for the attenuation coefficient. But you need to take into account also the scattering coefficient, especially for water, like if you want to see through the fog.

There is still a typing error on section title 4.2.1: between.
